# Discriminative Mixture-of-Experts on Graphs with Reliable Expert Fusion

**Haoyue Deng** [1]  **Menghui Wang** [1]  **Yunlong Zhou** [1]  **Ziwei Zhang** [2]  **Ran Zhang** [3]  **Chunming Hu** [1]  **Xiao Wang** [1] [*]

## Abstract

Graph Mixture-of-Experts (Graph-MoE) offers a way to scale GNNs via adaptive capacity allocation, with the goal of allowing different experts to capture diverse graph patterns. Its effectiveness heavily depends on the coordination between routing decisions and expert specialization. However, through extensive empirical study, we identify two critical phenomena. First, discrimination loss occurs on both the expert and routing sides, where GNN experts become highly homogenized and the router collapses to a small subset of experts, failing to reflect diverse graph semantics. Second, routing uncertainty is prevalent, as existing routers produce uncertain expert assignments for most nodes, and such uncertainty exhibits a strong negative correlation with model performance. To address these issues, we propose $C^2$GMoE, a novel **G**raph-**MoE** framework featuring **C**ontrastive routing and **C**onfidence-aware fusion. We introduce a group-wise contrastive routing strategy that provides explicit guidance for routing optimization by aligning node-level routing decisions with semantic clusters while satisfying load-balancing constraints. Moreover, through a theoretical analysis of generalization error, we develop a confidence-aware fusion mechanism that adaptively reweights expert predictions according to their confidence. Extensive experiments across multiple benchmarks demonstrate the effectiveness of our proposed $C^2$GMoE. Code is publicly available at https://github.com/Gatlin1111/C2GMoE.

## 1. Introduction

Graph Neural Networks (GNNs) (Kipf & Welling, 2017; Velickovic et al., 2018; Hamilton et al., 2017) have emerged as a standard approach to learn high-quality representations from graph data (Zhang et al., 2022; Wu et al., 2022), using message-passing to combine graph structures and node features. However, traditional GNNs face a significant bottleneck in terms of model capacity (Zeng et al., 2021). Specifically, increasing model depth often leads to over-smoothing (Li et al., 2018; Oono & Suzuki, 2020; Rong et al., 2020) and over-squashing problems (Topping et al., 2022; Deac et al., 2022), while simply increasing model width significantly raises computational costs. Moreover, real-world graphs often contain diverse local patterns and heterogeneous information (Zhu et al., 2021), which are difficult to capture using the unified parameter space typical of traditional GNNs (Wang et al., 2023; Zeng et al., 2024).

To address these limitations, recent studies have introduced mixture-of-experts (MoE) (Shazeer et al., 2017; Fedus et al., 2022; Lepikhin et al., 2021) architectures into graph learning to adaptively allocate model capacity and capture diverse patterns (Wang et al., 2023; Zeng et al., 2024; Yao et al., 2025; Shi et al., 2025; Chen et al., 2025). A typical Graph-MoE framework mainly comprises a gating network and a set of expert models. The gating network acts as a router that dynamically assigns each node to a subset of experts. By activating only a subset of experts per node, this architecture enables the model to scale its capacity while maintaining inference efficiency. Each expert is trained to capture distinct patterns of the graph data, potentially with distinct inductive biases, and the outputs of the selected experts are aggregated according to the routing weights.

However, despite their potentials, existing Graph-MoE models are far from perfect. Through extensive empirical study (Section 2), we reveal two critical bottlenecks:

- **Discrimination Loss:** Ideally, different GNN experts should capture diverse and complementary graph patterns, while the routing mechanism should precisely distinguish among these experts to assign nodes to their most compatible GNN experts. However, our empirical findings reveal a severe loss of discrimination across both sides. On the expert side, we observe pervasive

---

[1]School of Software, Beihang University, Beijing, China [2]School of Computer Science and Engineering, Beihang University, Beijing, China [3]China CITIC Bank, Beijing, China. Correspondence to: Xiao Wang <xiao_wang@buaa.edu.cn>.

*Proceedings of the 43rd International Conference on Machine Learning*, Seoul, South Korea. PMLR 306, 2026. Copyright 2026 by the author(s).

expert homogenization, where different GNN experts converge to highly redundant and overlapping representation spaces, losing their functional distinctiveness. On the routing side, the gating network suffers from routing collapse, failing to differentiate between experts and collapsing to a small subset of GNN experts regardless of the nodes' structural roles and neighborhood contexts.

- **Routing Uncertainty:** A reliable routing mechanism should be capable of making confident expert assignments by clearly identifying the GNN expert that is most compatible with a given node. However, our empirical analysis shows that existing models frequently produce highly uncertain decisions, assigning uniform weights to multiple experts. Moreover, we find that this routing ambiguity is strongly correlated with performance degradation: nodes associated with high routing entropy consistently exhibit lower classification accuracy. Such uncertainty indicates that the router fails to reliably identify and select GNN experts that are matched to a node's specific structural characteristics or attribute patterns.

To address these issues, we propose C²GMoE, a novel **G**raph **M**ixture-**o**f-**E**xperts framework with **C**ontrastive Routing Learning and **C**onfidence-aware Fusion. To mitigate discrimination loss, we introduce a group-wise contrastive routing strategy. This strategy partitions nodes into semantic groups and provides explicit guidance for routing optimization by aligning node-level routing decisions with different graph attributes and topological structures. By applying contrastive routing objectives, it encourages different experts to capture distinct patterns, while satisfying load-balancing constraints. Furthermore, to address routing uncertainty, we conduct a theoretical analysis of the generalization error, proving that minimizing the error bound requires a positive correlation between fusion weights and expert predictive confidence. Based on theoretical insights, we further propose a theorem-guided confidence-aware fusion mechanism. This mechanism dynamically estimates the predictive confidence of each GNN expert and adaptively reweights their contributions, thereby reducing the impact of unreliable expert predictions.

We summarize our contributions as follows:

- We demonstrate two crucial phenomena in existing Graph-MoE methods: discrimination loss and routing uncertainty, indicating that existing Graph-MoE methods fail to fully realize the architectural potential due to insufficient discrimination among experts over diverse graph patterns and unreliable expert selection under uncertain routing.

- We propose C²GMoE, a novel Graph-MoE method that integrates group-wise contrastive routing to restore discriminative power by promoting expert specialization

under load-balancing constraints. Based on a theoretical analysis of the generalization error, we develop a theorem-guided confidence-aware fusion mechanism to adaptively and reliably aggregate expert predictions.

- Extensive experiments across multiple benchmark datasets demonstrate that C²GMoE consistently improves the discriminative power of experts and routings compared to existing Graph-MoE methods.

## 2. Empirical Study

**Notations.** Let $\mathcal{G} = (V, E)$ be a graph with $N$ nodes and $|E|$ edges, described by an adjacency matrix $\mathbf{A} \in \{0,1\}^{N \times N}$ and a feature matrix $\mathbf{X} \in \mathbb{R}^{N \times d}$. The $i$-th row of $\mathbf{X}$, denoted as $\mathbf{x}_i \in \mathbb{R}^d$, represents the initial feature vector of node $i$. The node labels are represented with a label matrix $\mathbf{Y} = [\mathbf{y}_u] \in \mathbb{R}^{N \times C}$, where $\mathbf{y}_u$ is the one-hot label of node $u$. We use bold uppercase letters to represent matrices and bold lowercase letters to represent vectors.

**Observation 1 (Discrimination Loss).** To assess whether existing Graph-MoE models successfully achieve discriminative specialization—where GNN experts capture diverse graph patterns and the router precisely distinguishes among them, we analyze GMoE (Wang et al., 2023) as a representative model across several benchmark datasets, including Photo (Shchur et al., 2018), Wikics (Mernyei & Cangea, 2020), and Flickr (Zeng et al., 2020). We employ a 2-layer GCN backbone with $M = 8$ experts and Top-2 gating, applying its default load-balancing loss with weight $\lambda = 0.1$, which is designed to encourage a uniform distribution of nodes among experts.

We first investigate the degree of expert specialization within the model. Specifically, after training, we feed all nodes into every expert individually. This results in a set of representation matrices. Then we visualize the Centered Kernel Alignment (CKA) similarity (Kornblith et al., 2019) between the representations produced by different experts. A higher CKA score indicates a stronger semantic alignment between two representation spaces, suggesting that the experts have captured similar representations. As illustrated in Figure 1, the similarity scores between different experts are significantly high across all datasets, with most pairs exceeding 0.8. This pervasive high-value pattern reveals a severe loss of discrimination among experts. Instead of capturing specialized graph patterns, the GCN experts converge toward a redundant and overlapping representation space.

Second, we investigate the expert utilization, which we define as the distribution of node assignments across different experts. As shown in Figure 2, where darker colors indicate a higher proportion of assigned nodes, we observe that a vast majority of nodes are assigned to a few dominant experts, while others remain largely under-utilized. This routing

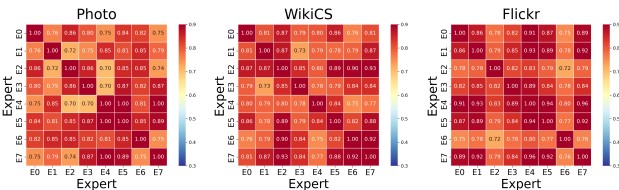

*Figure 1.* The CKA similarity between expert representations of GMoE (Wang et al., 2023) on three datasets.

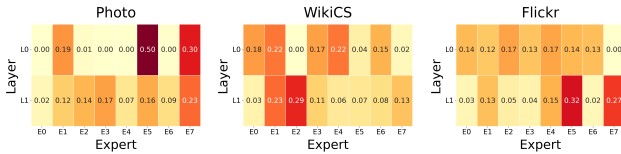

*Figure 2.* Expert load distribution of GMoE (Wang et al., 2023) on three datasets.

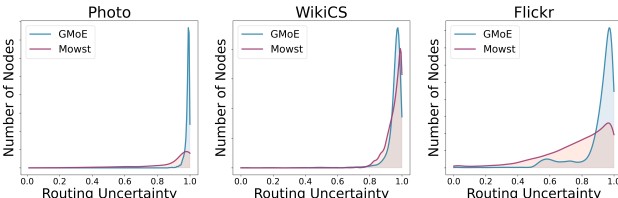

*Figure 3.* Distribution of routing uncertainty (entropy) across all nodes on GMoE (Wang et al., 2023) and Mowst (Zeng et al., 2024).

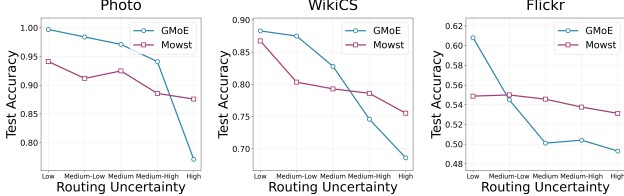

*Figure 4.* Classification accuracy versus routing uncertainty on GMoE (Wang et al., 2023) and Mowst (Zeng et al., 2024).

collapse confirms the loss of discrimination in the gating network. Even with a non-negligible penalty ($\lambda = 0.1$), the vanilla balancing mechanism fails to prevent routing collapse, thereby limiting the model's overall capacity.

**Observation 2 (Routing Uncertainty).** We observe that existing Graph-MoE models suffer from pervasive routing uncertainty, which significantly undermines model performance. Formally, for node i with representation $\mathbf{h}_i$, let $\mathbf{p}_i = [g_1(\mathbf{h}_i), g_2(\mathbf{h}_i), \ldots, g_M(\mathbf{h}_i)]$ denote the routing probabilities across $M$ experts, where $g_m(\mathbf{h}_i)$ is the gating weight for the $m$-th expert. The routing uncertainty is measured by the Shannon entropy: $H(\mathbf{p}_i) = -\sum_{m=1}^{M} g_m(\mathbf{h}_i) \log g_m(\mathbf{h}_i)$. High entropy indicates that the router is uncertain about which expert should handle the node.

We first examine the distribution of routing entropy across all nodes in Figure 3. Strikingly, for both GMoE (Wang et al., 2023) and Mowst (Zeng et al., 2024), we observe that a vast majority of nodes exhibit high routing uncertainty, suggesting that existing routing mechanisms frequently fail to produce confident expert assignments in practice.

To evaluate the impact of this uncertainty, we plot the node classification accuracy against routing entropy. As shown in Figure 4, we group nodes into five categories based on their routing entropy, ranging from Low to High. We observe a consistent and sharp decline in classification accuracy as routing uncertainty increases across all datasets. This indicates that the standard routing-based fusion mechanism is highly vulnerable to routing ambiguity: when the router is uncertain, the resulting weights fail to reflect the actual reliability of experts. This motivates us to move beyond simple weighted averaging and design a confidence-aware fusion mechanism that adaptively accounts for the confidence of each expert.

## 3. The Proposed Method

In this section, we propose C$^2$GMoE, a novel **G**raph **M**ixture-**o**f-**E**xperts framework with **C**ontrastive Routing Learning and **C**onfidence-aware Fusion. The overall framework is illustrated in Figure 5. Firstly, the graph is processes through a multi-layer Graph-MoE architecture (a). To ensure expert specialization and prevent routing collapse, we design a contrastive routing learning strategy (b), where a dual grouping module is incorporated to partition nodes from both attribute and structural perspectives, providing explicit guidance ($Loss_{rc}$) to align routing decisions with diverse graph patterns. To address routing uncertainty, we introduce a confidence-aware fusion mechanism (c) to adaptively integrate information across different experts. This mechanism addresses routing uncertainty by dynamically aggregating expert outputs based on predicted confidence scores, which is theoretically proven to reduce the error bound of model predictions.

### 3.1. The C$^2$GMoE Layer

We first define the C$^2$GMoE layer, the fundamental building block of our framework. The initial representation is given by $\mathbf{H}^{(0)} = \mathbf{X}$. The model consists of $L$ stacked MoE layers. In each layer, a node $i$ is processed by a shared expert *sh* and a set of $M$ specialized experts. The shared expert captures universal patterns across the graph, reducing redundancy and enabling the specialized experts to focus on distinct structural and attribute patterns. Specifically, the output of each expert is defined as:

$$\mathbf{h}_{i,m}^{(l)} = \sum_{j \in \mathcal{N}(i) \cup \{i\}} E_m\left(\mathbf{h}_j^{(l-1)}; \mathbf{W}_m^{(l)}\right), \quad (1)$$

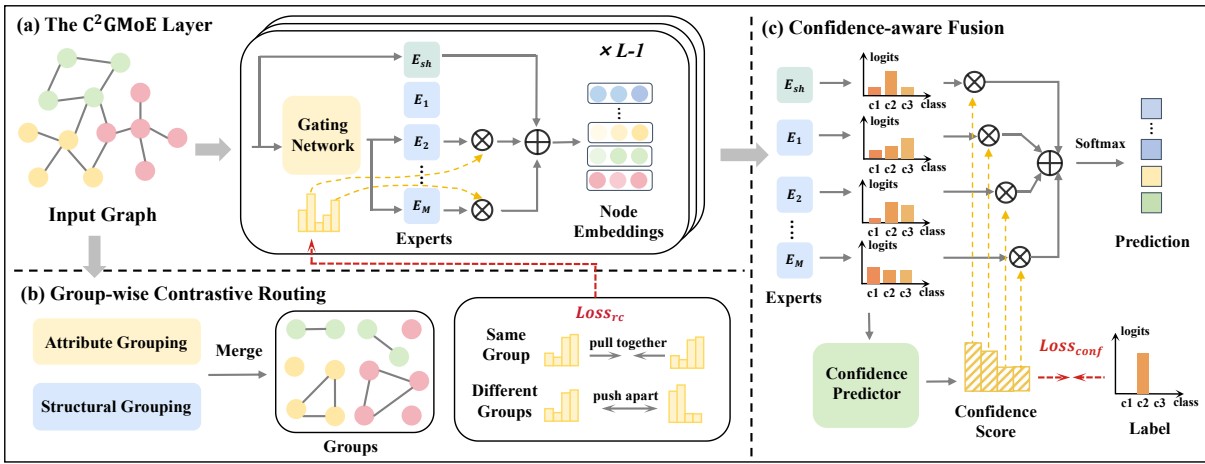

*Figure 5.* An overview of the proposed method: **(a) The C²GMoE Layer:** A sparse multi-layer architecture routing nodes to a shared expert ($E_{sh}$) and specialized experts ($E_1 \ldots E_M$) to generate embeddings. **(b) Group-wise Contrastive Routing**: A strategy using $\mathcal{L}_{rc}$ to align routing decisions with attribute and structural groups, enhancing expert specialization. **(c) Confidence-aware Fusion:** A mechanism reweights expert logits based on predicted confidence for reliable final prediction.

where $m \in \{\text{sh}, 1, \ldots, M\}$. Here, $\mathbf{h}_j^{(l-1)}$ denotes the input representation of node $j$ to layer $l$, and $E_m(\cdot)$ denotes a message-passing expert, which can be implemented by various GNN backbones such as GCN (Kipf & Welling, 2017) or GIN (Xu et al., 2019).

The specialized experts are selectively activated via a routing mechanism (Shazeer et al., 2017):

$$\mathbf{q}_i^{(l)} = \mathbf{h}_i^{(l-1)}\mathbf{W}_g + \epsilon \cdot \text{softplus}(\mathbf{h}_i^{(l-1)}\mathbf{W}_n), \quad (2)$$

where $\mathbf{W}_g$ and $\mathbf{W}_n$ are learnable projection matrices for routing scores and noise scales, respectively. The noise term $\epsilon$ is sampled from a standard normal distribution and introduces stochasticity into the routing process, helping the model avoid premature convergence to suboptimal expert assignments. The final gating weights $\mathbf{g}_i^{(l)} \in \mathbb{R}^M$ are:

$$\mathbf{g}_i^{(l)} = \text{softmax}(\text{TopK}(\mathbf{q}_i^{(l)}, k)), \quad (3)$$

where the TopK($\cdot$) operator keeps only the $k$ largest elements of the routing vector $\mathbf{q}_i$.

The node representation is then updated by aggregating the outputs from both types of experts:

$$\mathbf{h}_i^{(l)} = \mathbf{h}_{i,\text{sh}}^{(l)} + \sum_{m=1}^{M} g_{i,m}^{(l)} \cdot \mathbf{h}_{i,m}^{(l)}, \quad (4)$$

where $g_{i,m}^{(l)}$ is the $m$-th element of $\mathbf{g}_i^{(l)}$, representing the gating weight assigned to the $m$-th specialized expert for node $i$ at layer $l$.

### 3.2. Group-wise Contrastive Routing Strategy

To ensure effective expert specialization and prevent routing collapse, we propose a Group-wise Contrastive Routing

(GCR) strategy. We first partition nodes from both attribute and structural perspectives. *(1) Attribute grouping*: Nodes are clustered into $K$ clusters in the raw feature space $\mathbf{X}$ to capture attribute-level semantics. Each node $i$ is assigned an attribute cluster label. *(2) Structural grouping*: We utilize Personalized PageRank (PPR) to capture global topological roles. Each node is represented by its structural influence score, and these representations are clustered into $K$ clusters, yielding an structural cluster label.

The final $C$ disjoint semantic groups are determined by the joint labels from both perspectives. Specifically, two nodes $i$ and $j$ are assigned to the same semantic group if and only if they share identical labels in both attribute and structural views. This process ensures that each group consists of nodes with highly consistent feature patterns and topological roles.

For each semantic group $G_c$, we define the routing prototype $\mathbf{r}_c^{(l)} \in \mathbb{R}^M$ as the mean gating vector across all nodes in the group at layer $l$:

$$\mathbf{r}_c^{(l)} = \frac{1}{|G_c|} \sum_{i \in G_c} \mathbf{g}_i^{(l)}. \quad (5)$$

This prototype represents the aggregate expert preference of the $c$-th semantic group.

Then we introduce a routing contrastive loss $\mathcal{L}_{rc}$, which encourages the routing distribution $\mathbf{g}_i^{(l)}$ of each node to align with the prototype $\mathbf{r}_c^{(l)}$ of its assigned group while remaining distinguishable from other groups. Specifically, for each group $G_c$, we treat $(\mathbf{g}_i^{(l)}, \mathbf{r}_c^{(l)})$ as a positive pair for all $i \in G_c$, and $(\mathbf{g}_i^{(l)}, \mathbf{r}_{c'}^{(l)})$ with $c' \neq c$ as negative pairs.

The routing contrastive loss for layer $l$ is defined as:

$$\mathcal{L}_{\text{rc}}^{(l)} = -\sum_{c=1}^{C} \sum_{i \in G_c} \log \frac{\exp\left(\text{sim}(\mathbf{g}_i^{(l)}, \mathbf{r}_c^{(l)})/\tau\right)}{\sum_{c'=1}^{C} \exp\left(\text{sim}(\mathbf{g}_i^{(l)}, \mathbf{r}_{c'}^{(l)})/\tau\right)},$$
(6)

where $\text{sim}(\cdot, \cdot)$ denotes cosine similarity and $\tau$ is a temperature parameter. This design prevents routing collapse while avoiding uniform expert activation, thereby promoting specialization of experts across distinct semantic subspaces.

### 3.3. Confidence-aware Fusion Mechanism

To address the performance degradation caused by routing uncertainty and adaptively integrate information from diverse experts, we propose the Theorem-guided Confidence-aware Fusion (CAF) mechanism. We revisit the generalization error bound of the Graph MoE Model and establish a theoretically grounded fusion mechanism that dynamically adapts to expert reliability (Cao et al., 2024).

**Theoretical Analysis.** We investigate the generalization error (GE) of our proposed model. In an $L$-layer architecture, each node $i$ traverses a sequence of experts, forming a specific path $p$ from the set of all potential paths $\mathcal{P}$ (where $|\mathcal{P}| = (1 + M)^L$). We define the path-level weight as $w_{i,p} = \prod_{l=1}^{L} g_{i,m_p^{(l)}}^{(l)}$, where $m_p^{(l)}$ is the index of the expert at layer $l$ along path $p$. Let $\hat{\mathbf{y}}_{i,p}$ denote the path-specific prediction for node $i$, and the corresponding path loss be $\mathcal{L}_{i,p} = \ell(\hat{\mathbf{y}}_{i,p}, y_i)$. The generalization error of the model $f$ over the unknown joint distribution $\mathcal{D}$ is defined as $\text{GE}(f) = \mathbb{E}_{(\mathbf{x},y)\sim\mathcal{D}}[\ell(f(\mathbf{x}), y)]$.

**Theorem 3.1** (Generalization Error Bound). *Let $\hat{e}(\mathbf{f}_p)$ denote the empirical error of path $p \in \mathcal{P}$ on the training set. With a confidence level of $1 - \Delta$, the generalization error of the model satisfies:*

$$GE(f) \leq \sum_{p \in \mathcal{P}} \hat{e}(\mathbf{f}_p) + \sum_{p \in \mathcal{P}} \left[ \frac{1}{|\mathcal{P}|} Cov(w_p, \mathcal{L}_p) - \frac{|\mathcal{P}| - 1}{|\mathcal{P}|} \sum_{q \neq p} Cov(w_p, \mathcal{L}_q) \right] + \mathcal{R}(|\mathcal{P}|, N, \Delta),$$
(7)

*where $Cov(\cdot, \cdot)$ denotes the covariance, $\mathcal{R}$ denotes the term associated with Rademacher complexity.*

The proof of Theorem 3.1 is provided in Appendix B.1. Note that $\mathcal{R}$ and the empirical error $\hat{e}(\mathbf{f}_p)$ remain constant when optimizing the weights for a given model. The bound reveals that the generalization ability of our model is significantly influenced by the interaction between routing weights and path losses. To minimize the upper bound, the following principles should be satisfied: (1) a negative correlation between a path's weight and its own loss ($\text{Cov}(w_p, \mathcal{L}_p) < 0$), and (2) a positive correlation between a path's weight and the losses of alternative paths ($\text{Cov}(w_p, \mathcal{L}_q) > 0$).

However, directly employing path loss $\mathcal{L}_p$ to formulate fusion weights is impractical. During the optimization process, $\mathcal{L}_p$ is minimized toward zero, and its unbounded range $[0, +\infty)$ makes it highly sensitive to small perturbations, leading to numerical instability. To address this, we consider the cross-entropy loss $\mathcal{L}_p = -\sum_k y_k \log p_k$, where $y_k$ is the one-hot label vector, and $p_k$ represents the predicted probability for the k-th class. Given the ground-truth label, this simplifies to $\mathcal{L}_p = -\log p_{\text{true}}$, where $p_{\text{true}} \in [0, 1]$ denotes the probability assigned to the ground-truth class.

Since $\mathcal{L}_p$ is strictly monotonically decreasing with respect to $p_{\text{true}}$, we use $p_{\text{true}}$ as a stable confidence proxy. This translates the requirements of Theorem 3.1 into the following actionable corollaries for weight design:

**Corollary 3.2.** *To minimize the generalization error bound, a positive correlation should exist between a path's weight $w_p$ and its correctness probability $p_{true,p}$.*

**Corollary 3.3.** *A negative correlation is desirable between a path's weight $w_p$ and the correctness probabilities of alternative paths $p_{true,q}$, for all $q \neq p$.*

The proofs are provided in Appendix B.2 and B.3. Above analysis motivates designing weights based on a confidence measure that reflects the correctness probability.

**Confidence Prediction and Fusion.** While Theorem 3.1 provides a global optimization perspective for the entire model, a multi-layer implementation of confidence-aware fusion would exponentially increase computational complexity, contradicting the principle of sparse activation in MoE. To maintain computational tractability while retaining the benefits of theoretical guidance, we focus on the final layer as a centralized decision-making module.

For each node $i$ in the final layer, we consider the experts $\mathcal{M}^+ = \{\text{sh}, 1, \ldots, M\}$. Let $\mathbf{z}_{i,m} \in \mathbb{R}^C$ denote the representation output by the $m$-th expert ($m \in \mathcal{M}^+$). To satisfy Corollary 3.2, we first compute the individual expert confidence $c_{i,m}^{\text{ind}}$ as a proxy for the correctness probability:

$$c_{i,m}^{\text{ind}} = \sigma(\text{Predictor}(\mathbf{z}_{i,m})), \quad \forall m \in \mathcal{M}^+,$$
(8)

where $\sigma(\cdot)$ is the sigmoid function.

To fulfill the requirement of cross-expert interaction derived from Corollary 3.3, we further define a relative confidence $c_{i,m}^{\text{rel}}$ to represent the comparative reliability of expert $m$:

$$c_{i,m}^{\text{rel}} = \frac{\sum_{j \in \mathcal{M}^+, j \neq m} \log c_{i,j}^{\text{ind}}}{\sum_{k \in \mathcal{M}^+} \log c_{i,k}^{\text{ind}} + \xi}.$$
(9)

To prevent potential division-by-zero issues, we apply a small smoothing factor $\xi > 0$ when computing relative confidence. Since $\log c_{i,j}^{\text{ind}} < 0$, a lower confidence in alternative experts ($j \neq m$) leads to a more negative numerator, thereby increasing $c_{i,m}^{\text{rel}}$ and boosting the weight of expert $m$.

The total confidence score is defined as $c_{i,m} = c_{i,m}^{\text{ind}} + c_{i,m}^{\text{rel}}$. To obtain the final fusion weights $w_{i,m}$, we apply a softmax normalization over all experts in $\mathcal{M}^+$:

$$w_{i,m} = \frac{\exp(c_{i,m})}{\sum_{q \in \mathcal{M}^+} \exp(c_{i,q})}. \tag{10}$$

Finally, the comprehensive node representation $\mathbf{z}_i$ is formed by the weighted aggregation of all expert outputs:

$$\mathbf{z}_i = \sum_{m \in \mathcal{M}^+} w_{i,m} \cdot \mathbf{z}_{i,m}. \tag{11}$$

The final prediction is $\hat{\mathbf{y}}_i = \text{Softmax}(\mathbf{z}_i)$.

To ensure that the individual confidence $c_{i,m}^{\text{ind}}$ accurately estimates the expert's confidence, we optimize the predictor to approximate the correctness probability. This target, denoted as $p_{i,m}^{\text{true}}$, is defined as the softmax score assigned by the $m$-th expert to the ground-truth class $y_i$, i.e., $p_{i,m}^{\text{true}} = \text{Softmax}(\mathbf{z}_{i,m})_{y_i}$. The predictor is then trained via a Mean Absolute Error (MAE) loss:

$$\mathcal{L}_{\text{conf}} = \frac{1}{N \cdot |\mathcal{M}^+|} \sum_{i=1}^{N} \sum_{m \in \mathcal{M}^+} |c_{i,m}^{\text{ind}} - p_{i,m}^{\text{true}}|. \tag{12}$$

### 3.4. Objective Function

The entire framework is trained end-to-end by minimizing a joint objective that combines the task-specific loss, the routing contrastive loss, and the confidence loss:

$$\mathcal{L} = \mathcal{L}_{\text{task}} + \alpha \mathcal{L}_{\text{rc}} + \gamma \mathcal{L}_{\text{conf}}, \tag{13}$$

where $\alpha$ and $\gamma$ are hyperparameters controlling the relative importance of the auxiliary losses. Specifically, $\mathcal{L}_{\text{task}}$ is the standard cross-entropy loss for node classification. The routing contrastive loss $\mathcal{L}_{\text{rc}}$ is defined as the average across the first $L - 1$ layers: $\mathcal{L}_{\text{rc}} = \frac{1}{L-1} \sum_{l=1}^{L-1} \mathcal{L}_{\text{rc}}^{(l)}$.

## 4. Experiments

**Datasets.** For the benchmarks, we choose six widely used graph datasets: Photo, Computers, CS (Shchur et al., 2018), WikiCS (Mernyei & Cangea, 2020), Flickr (Zeng et al., 2020), and Ogbn-arxiv (Hu et al., 2020). The detailed statistics of the datasets can be found in Appendix C. We select Micro-F1 and Macro-F1 as metrics to conduct a comprehensive performance comparison.

**Baselines and Implementations.** We compare our proposed method against two categories of methods: (1) Standard GNNs, including GCN (Kipf & Welling, 2017), Graph-SAGE (Hamilton et al., 2017), and GIN (Xu et al., 2019), which are widely used and representative GNN architectures, and (2) Graph-MoE methods, including GMoE (Wang

et al., 2023), Mowst (Zeng et al., 2024), and DA-MoE (Yao et al., 2025). Descriptions of the baselines are provided in Appendix D. Since Graph-MoE methods can be combined with different expert networks, we utilize the three standard GNNs as the expert backbone for both Graph-MoE baselines and our C²GMoE. To ensure a fair comparison with comparable FLOPs, we scale the hidden dimensions (Wang et al., 2023); the detailed complexity analysis is provided in Appendix E. Additional implementation details are provided in Appendix F. All results are averaged over 10 runs.

### 4.1. Node Classification Results

Experimental results for node classification across six benchmarks are summarized in Tables 1 and 2, with the performance using GIN experts reported in Appendix G.1. The bold and underline markers denotes the best and second-best performance respectively. We have the following observations: (1) C²GMoE consistently improves the performance of all baseline models across all datasets, demonstrating the effectiveness of the proposed method. (2) Across different expert architectures, our method shows stable and consistent improvements, indicating that the approach is largely independent of the choice of the underlying expert network. (3) The gains in Macro-F1 are generally more pronounced than those in Micro-F1, suggesting that our method better captures and balances performance for minority classes.

### 4.2. Ablation Studies & Analysis

**Ablation Study.** To evaluate the effectiveness of our method, we conduct ablation studies by removing two fundamental components of our method: the Group-wise Contrastive Routing Strategy (denoted as w/o Route) and the Confidence-Aware Fusion mechanism (denoted as w/o Conf). The results are summarized in Table 3. The full model consistently achieves the best performance across all metrics and datasets. Removing the GCR strategy consistently degrades both Micro-F1 and Macro-F1, highlighting its importance in guiding experts to specialize on semantically coherent groups. Disabling the CAF mechanism also leads to notable performance drops, especially in Macro-F1, indicating that adaptively weighting experts based on their reliability is critical.

**Expert Load Analysis.** To validate the effectiveness of our proposed GCR strategy in promoting balanced expert utilization, we visualize the expert load distribution across two datasets, WikiCS and Flickr, in Figure 6. We compare three settings: (1) "No Loss", where the model is trained without any auxiliary routing constraints or our GCR strategy; (2) "Balanced Loss", where our GCR strategy is replaced by the standard load-balancing loss; (3) "Ours", which employs the proposed GCR strategy to guide expert assignment. The vertical axis represents the proportion of nodes assigned to

*Table 1.* Performance comparison of different methods with GCN experts (%).

| Dataset | Metric | GCN | GMoE-GCN | Mowst-GCN | DAMoE-GCN | $C^2$GMoE-GCN |
|---|---|---|---|---|---|---|
| Photo | Micro-F1 | 94.25 ± 0.59 | 94.55 ± 0.30 | 94.37 ± 0.34 | 94.14 ± 0.40 | **95.16 ± 0.20** |
| | Macro-F1 | 93.52 ± 0.72 | 93.83 ± 0.36 | 93.89 ± 0.36 | 93.86 ± 0.53 | **94.64 ± 0.26** |
| WikiCS | Micro-F1 | 78.87 ± 0.60 | 79.47 ± 0.83 | 79.73 ± 0.73 | 79.12 ± 0.70 | **80.11 ± 0.63** |
| | Macro-F1 | 76.08 ± 0.94 | 76.86 ± 1.02 | 77.07 ± 0.90 | 76.14 ± 1.13 | **77.47 ± 0.87** |
| Computers | Micro-F1 | 90.37 ± 0.62 | 91.75 ± 0.48 | 86.19 ± 1.23 | 91.17 ± 0.44 | **92.88 ± 0.18** |
| | Macro-F1 | 89.09 ± 1.02 | 90.96 ± 0.82 | 84.87 ± 1.02 | 90.19 ± 0.58 | **92.27 ± 0.24** |
| CS | Micro-F1 | 94.10 ± 0.41 | 94.21 ± 0.44 | 93.34 ± 0.38 | 93.41 ± 0.13 | **94.82 ± 0.24** |
| | Macro-F1 | 92.60 ± 0.66 | 92.84 ± 0.56 | 91.07 ± 0.36 | 91.65 ± 0.15 | **93.43 ± 0.41** |
| Flickr | Micro-F1 | 53.67 ± 0.13 | 54.70 ± 0.21 | 54.56 ± 0.48 | 53.84 ± 0.72 | **55.01 ± 0.48** |
| | Macro-F1 | 24.80 ± 0.40 | 25.22 ± 0.87 | 25.36 ± 0.55 | 24.31 ± 1.15 | **26.60 ± 0.55** |
| ogbn-arxiv | Micro-F1 | 71.61 ± 0.24 | 71.93 ± 0.21 | 72.54 ± 0.14 | 71.95 ± 0.17 | **72.74 ± 0.15** |
| | Macro-F1 | 50.81 ± 0.30 | 51.03 ± 0.28 | 52.59 ± 0.25 | 51.48 ± 0.32 | **52.72 ± 0.18** |

*Table 2.* Performance comparison of different methods with SAGE experts (%).

| Dataset | Metric | SAGE | GMoE-SAGE | Mowst-SAGE | DAMoE-SAGE | $C^2$GMoE-SAGE |
|---|---|---|---|---|---|---|
| Photo | Micro-F1 | 95.03 ± 0.57 | 95.35 ± 0.42 | 95.25 ± 0.35 | 95.71 ± 0.19 | **96.19 ± 0.14** |
| | Macro-F1 | 93.98 ± 0.80 | 94.58 ± 0.49 | 94.24 ± 0.54 | 95.14 ± 0.24 | **95.45 ± 0.21** |
| WikiCS | Micro-F1 | 78.76 ± 0.44 | 79.14 ± 0.55 | 78.94 ± 0.62 | 78.71 ± 0.82 | **80.09 ± 0.53** |
| | Macro-F1 | 76.07 ± 0.57 | 76.32 ± 0.74 | 76.56 ± 0.72 | 76.28 ± 0.90 | **77.77 ± 0.38** |
| Computers | Micro-F1 | 91.40 ± 0.26 | 91.48 ± 0.71 | 91.75 ± 0.48 | 91.17 ± 0.26 | **92.37 ± 0.18** |
| | Macro-F1 | 90.26 ± 0.34 | 90.22 ± 1.04 | 90.44 ± 0.76 | 89.73 ± 0.22 | **91.70 ± 0.30** |
| CS | Micro-F1 | 94.69 ± 0.42 | 95.24 ± 0.26 | 95.00 ± 0.21 | 93.59 ± 0.11 | **95.99 ± 0.11** |
| | Macro-F1 | 93.37 ± 0.53 | 93.93 ± 0.37 | 93.17 ± 0.20 | 91.74 ± 0.23 | **94.82 ± 0.14** |
| Flickr | Micro-F1 | 53.24 ± 0.14 | 53.46 ± 0.22 | 53.68 ± 0.17 | 52.68 ± 0.28 | **54.12 ± 0.30** |
| | Macro-F1 | 22.79 ± 0.97 | 22.15 ± 0.81 | 23.95 ± 0.73 | 21.84 ± 0.54 | **25.13 ± 0.66** |
| ogbn-arxiv | Micro-F1 | 71.83 ± 0.06 | 71.53 ± 0.32 | 71.86 ± 0.26 | 71.78 ± 0.25 | **72.24 ± 0.20** |
| | Macro-F1 | 51.77 ± 0.23 | 50.08 ± 0.25 | 51.91 ± 0.29 | 51.44 ± 0.59 | **52.18 ± 0.04** |

*Table 3.* The results of ablation study for two key components.

| Dataset | Metric | w/o conf | w/o route | Full Model |
|---|---|---|---|---|
| Photo | Micro-F1 | 94.84 ± 0.19 | 94.92 ± 0.13 | **95.16 ± 0.20** |
| | Macro-F1 | 94.36 ± 0.13 | 94.43 ± 0.23 | **94.64 ± 0.26** |
| WikiCS | Micro-F1 | 79.81 ± 0.57 | 79.65 ± 0.52 | **80.11 ± 0.63** |
| | Macro-F1 | 76.88 ± 0.73 | 76.94 ± 0.68 | **77.47 ± 0.87** |
| Flickr | Micro-F1 | 54.82 ± 0.25 | 54.56 ± 0.17 | **55.01 ± 0.48** |
| | Macro-F1 | 26.24 ± 1.55 | 24.99 ± 1.03 | **26.60 ± 0.55** |
| ogbn-arxiv | Micro-F1 | 72.57 ± 0.28 | 72.64 ± 0.26 | **72.74 ± 0.15** |
| | Macro-F1 | 51.89 ± 0.20 | 52.53 ± 0.43 | **52.72 ± 0.18** |

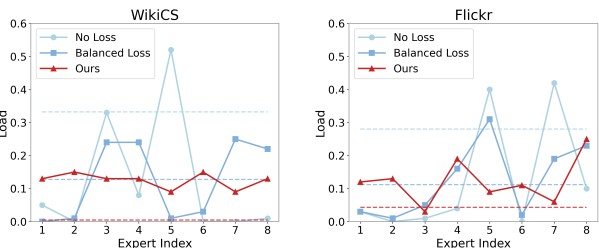

*Figure 6.* Comparison of expert load distribution.

each expert. Solid lines denote the mean load assigned to each expert, while dashed lines indicate the corresponding load variance. As shown, the "No Loss" setting suffers from severe expert collapse. The "Balanced Loss" attempts to mitigate this, but the load is still highly sporadic and unevenly distributed. In contrast, our proposed method achieves a uniform load distribution across all 8 experts, with the dashed red line showing markedly lower variance than the other two settings. These results demonstrate that the GCR strategy

effectively prevents expert collapse by explicitly aligning routing decisions with inherent group patterns.

**Expert Diversity Analysis.** To evaluate whether our GCR strategy facilitates functional specialization, we conduct both qualitative and quantitative analyses. First, we visualize the output embeddings from eight experts on the WikiCS and Flickr datasets using T-SNE dimensionality reduction. As illustrated in Figure 7, the node representations processed by different experts form distinct, well-separated

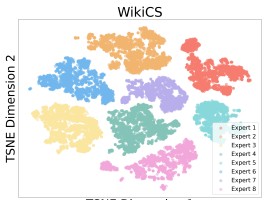 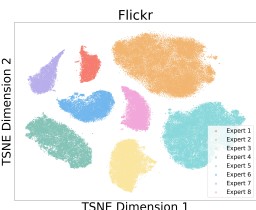

*Figure 7.* T-SNE visualization of expert output embeddings.

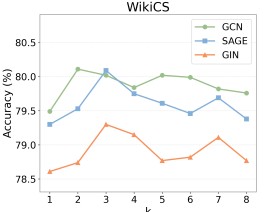 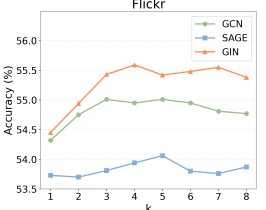

*Figure 9.* The parameter sensitivity of $C^2$GMoE for different $k$.

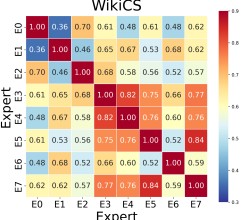 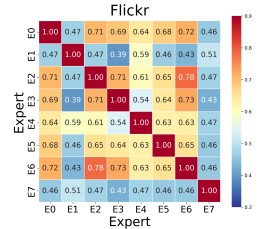

*Figure 8.* CKA similarity between expert representations. Low values indicate high expert diversity and functional specialization.

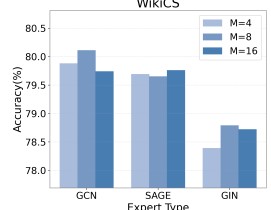 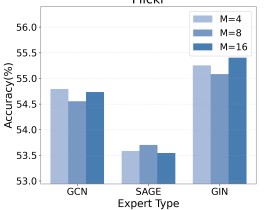

*Figure 10.* The parameter sensitivity of $C^2$GMoE for different $M$.

clusters without overlap. The clear boundary between clusters confirms that our mechanism effectively drives each expert to specialize in a unique subset graph patterns, rather than collapsing into redundant functional states. Second, we examine the CKA similarity between the outputs of different experts to measure feature redundancy. In contrast to the high similarity observed in GMoE (see Figure 1), the CKA heatmaps for our method in Figure 8 exhibit significantly lower similarity scores. Together, these results provide strong evidence that our strategy prevents functional homogenization, enabling experts to capture diverse and specialized semantic information across the graph.

**Parameter Study.** We analyze the key parameters: the number of active experts ($k$) and the total number of experts ($M$). First, We analyze the influence of the sparsity level by varying $k$ across $\{1, 2, \ldots, 8\}$ for different GNN backbones, while fixing the total expert count $M = 8$. The results shown in figure 9 indicate that optimal performance for the WikiCS dataset is achieved around $k = 2$ or $k = 3$, suggesting that a high degree of sparsity is most effective for this benchmark. Conversely, on the more complex Flickr dataset, performance generally improves as $k$ increases, peaking around $k = 4$ or $k = 5$, which implies that larger graphs benefit from activating more experts to adequately cover the feature space. Next, we analyze the sensitivity to the total expert count $M \in \{4, 8, 16\}$, while fixing the active expert count $k = 2$. The Results, summarized in figure 10, suggest that $C^2$GMoE exhibits strong robustness to the choice of $M$, effectively utilizing the increased capacity without incurring significant performance degradation. Furthermore, comprehensive sensitivity analyses for other auxiliary hyperparameters are provided in Appendix G.2.

## 5. Related Work

**Graph Neural Networks.** Graph Neural Networks (GNNs) have achieved significant success in graph learning by leveraging message-passing mechanisms, such as GCN (Kipf & Welling, 2017), GAT (Velickovic et al., 2018), and Graph-SAGE (Hamilton et al., 2017). Recent efforts have introduced various refinements to enhance their capability. To address graph heterophily, GPRGNN (Chien et al., 2021) and FAGCN (Bo et al., 2021) focus on handling graph heterophily via adaptive filtering and signal decoupling. To enable deeper architectures, techniques like residual connections (He et al., 2016) and normalization (Ba et al., 2016; Ioffe & Szegedy, 2015) have been incorporated. However, most GNNs rely on a static, shared parameter space designed for global graph properties, limiting their ability to adapt to the complex local patterns of real-world graphs.

**Graph Mixture of Experts.** Originating from (Jacobs et al., 1991; Jordan & Jacobs, 1994; Chen et al., 1999), Mixture of Experts (MoE) has been popularized by Sparse MoE (Shazeer et al., 2017) for efficient model scaling, serving as a foundation for modern large models (Lepikhin et al., 2021; Fedus et al., 2022; Dai et al., 2024; Jiang et al., 2024). MoE has gained significant attention in graph learning. Early works applied MoE to graph-level tasks (Hu et al., 2022), later expanding to node classification (Han et al., 2024; Shi et al., 2025), link prediction (Ma et al., 2024), and OOD generalization (Wu et al., 2024). GMoE (Wang et al., 2023) introduces sparse top-$K$ gating into GNNs, enabling nodes to activate different experts. Mowst (Zeng et al., 2024) combines MLP and GNN experts, decoupling attribute and structural patterns. More recently, GNNMoE (Chen et al., 2025) integrates fine-grained message-passing operators, and Moscat (Deng et al., 2024) improves deeper GNN gen-

eralization via a mixture of scope experts at inference time. Despite these advances, existing methods suffer from the loss of discriminative power and high decision uncertainty.

## 6. Conclusion

In this paper, we identify two critical phenomena in Graph-MoE: discrimination loss and routing uncertainty. To address these, we propose $C^2$GMoE, with a Contrastive Routing strategy to guide routing decisions toward semantically meaningful graph patterns, and a Confidence-aware Fusion mechanism to calibrate expert contributions according to predictive confidence. Extensive experiments verify the effectiveness of $C^2$GMoE.

## Acknowledgements

This work was supported by the National Natural Science Foundation of China under Grant No. 62322203.

## Impact Statement

This paper presents work whose goal is to advance the field of Graph Machine Learning and will promote the application of graph machine learning in graph data mining, such as enhancing recommendation systems and social network analysis. There are many potential societal consequences of our work, none of which we feel must be specifically highlighted here.

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

## A. Preliminaries

**Graph Neural Networks.** Taking the classical Graph Convolutional Network (GCN) (Kipf & Welling, 2017) as a representative GNN, the propagation rule at the $l$-th layer is given by

$$\mathbf{h}_i^{(l+1)} = \sigma \left( \sum_{j \in \mathcal{N}(i) \cup \{i\}} \frac{1}{\sqrt{\tilde{d}_i \tilde{d}_j}} \mathbf{h}_j^{(l)} \mathbf{W}^{(l)} \right), \tag{14}$$

where $\mathbf{h}_i^{(l)}$ is the feature vector of node $i$ at the $l$-th layer, with $\mathbf{h}_i^{(0)} = \mathbf{x}_i$. $\mathcal{N}(i)$ denotes the set of neighbors of node $i$. $\tilde{d}_i$ represents the degree of node $i$ in the graph with self-loops. The aggregated feature is normalized by $\sqrt{\tilde{d}_i \tilde{d}_j}$. $\mathbf{W}^{(l)}$ is a trainable weight matrix, and $\sigma(\cdot)$ denotes a non-linear activation function.

**Mixture of Experts.** The MoE (Jacobs et al., 1991; Shazeer et al., 2017) model consists of a set of $M$ expert networks $\mathcal{E} = \{E_1, E_2, \ldots, E_M\}$ and a gating network $G$. Given input features $\mathbf{X}$, each expert produces a representation $E_i(\mathbf{X})$, while the gating network computes a routing vector $G(\mathbf{X}) \in \mathbb{R}^M$ over the experts. These scores are normalized to obtain gating weights, which are used to aggregate the expert outputs into the final representation:

$$\mathbf{H}_{\text{out}} = \sum_{i=1}^{M} G(\mathbf{X})_i \cdot E_i(\mathbf{X}). \tag{15}$$

## B. Proofs

### B.1. Proof of Theorem 3.1

Consider an $L$-layer model. For a node $i$, the output is recursively computed as

$$f(\mathbf{x}_i) = \sum_{e_1=1}^{M} g_{i,e_1}^{(1)} E_{i,e_1}^{(1)} \left( \sum_{e_2=1}^{M} g_{i,e_2}^{(2)} E_{i,e_2}^{(2)} \left( \cdots \sum_{e_L=1}^{M} g_{i,e_L}^{(L)} E_{i,e_L}^{(L)} (\mathbf{x}_i) \cdots \right) \right), \tag{16}$$

where $g_{i,e_l}^{(l)}$ is the gating weight for expert $e_l$ at layer $l$, and $E_{i,e_l}^{(l)}$ denotes the output of the corresponding expert.

Given the model output, the loss for node $i$ can be written as

$$\ell(f(\mathbf{x}_i), y_i) = \ell \left( \sum_{e_1=1}^{M} g_{i,e_1}^{(1)} E_{i,e_1}^{(1)} \left( \sum_{e_2=1}^{M} g_{i,e_2}^{(2)} E_{i,e_2}^{(2)} \left( \cdots \sum_{e_L=1}^{M} g_{i,e_L}^{(L)} E_{i,e_L}^{(L)} (\mathbf{x}_i) \cdots \right) \right), y_i \right). \tag{17}$$

Consider $\ell(\cdot, y)$ to be the convex logistic loss function. By Jensen's inequality, we first extract the outermost gating weights:

$$\ell(f(\mathbf{x}_i), y_i) \leq \sum_{e_1=1}^{M} g_{i,e_1}^{(1)} \ell \left( E_{i,e_1}^{(1)} \left( \sum_{e_2=1}^{M} g_{i,e_2}^{(2)} E_{i,e_2}^{(2)} (\cdots) \right), y_i \right). \tag{18}$$

We consider a simplified setting where each expert function $E_{i,e_l}^{(l)}$ is treated as a linear or non-decreasing convex transformation. Under this assumption, Jensen's inequality can be recursively applied to the inner aggregations, yielding

$$\ell(f(\mathbf{x}_i), y_i) \leq \sum_{e_1=1}^{M} \cdots \sum_{e_L=1}^{M} \left( \prod_{l=1}^{L} g_{i,e_l}^{(l)} \right) \ell(\hat{\mathbf{y}}_{i,p}, y_i), \tag{19}$$

where $p = (e_1, \ldots, e_L)$ denotes a specific expert path and $\hat{\mathbf{y}}_{i,p}$ is the corresponding path-level prediction.

By defining the path-level weight as $w_{i,p} = \prod_{l=1}^{L} g_{i,e_l}^{(l)}$ and the path loss as $\mathcal{L}_{i,p} = \ell(\hat{\mathbf{y}}_{i,p}, y_i)$, we obtain

$$\ell(f(\mathbf{x}_i), y_i) \leq \sum_{p \in \mathcal{P}} w_{i,p} \mathcal{L}_{i,p}. \tag{20}$$

Taking expectation over $(\mathbf{x}, y) \sim \mathcal{D}$ gives:

$$\mathrm{GE}(f) = \mathbb{E}[\ell(f(\mathbf{x}), y)] \leq \sum_{p \in \mathcal{P}} \mathbb{E}[w_p \mathcal{L}_p]. \tag{21}$$

Let $|\mathcal{P}|$ be the total number of paths. Then

$$
\begin{aligned}
\mathrm{GE}(f) &\leq \sum_{p \in \mathcal{P}} \mathbb{E}[w_p \mathcal{L}_p] \\
&= \frac{1}{|\mathcal{P}|} \sum_{p \in \mathcal{P}} \left( |\mathcal{P}| \, \mathbb{E}[w_p \mathcal{L}_p] \right) \\
&= \frac{1}{|\mathcal{P}|} \sum_{p \in \mathcal{P}} \left( \mathbb{E}[w_p \mathcal{L}_p] + (|\mathcal{P}| - 1) \, \mathbb{E}\Big[(1 - \sum_{q \neq p} w_q) \mathcal{L}_p\Big] \right)
\end{aligned}
$$

Then, we expand each expectation of a product using $\mathbb{E}[AB] = \mathbb{E}[A]\mathbb{E}[B] + \mathrm{Cov}(A, B)$. Specifically, $\mathbb{E}[w_p \mathcal{L}_p]$ and $\mathbb{E}[w_q \mathcal{L}_p]$ are decomposed into expectation and covariance terms, while the linearity of expectation is applied to the summation over $q \neq p$. Then, we have:

$$
\begin{aligned}
\mathrm{GE}(f) &\leq \frac{1}{|\mathcal{P}|} \sum_{p \in \mathcal{P}} \left( \mathbb{E}[w_p]\mathbb{E}[\mathcal{L}_p] + \mathrm{Cov}(w_p, \mathcal{L}_p) - (|\mathcal{P}| - 1) \sum_{q \neq p} \big( \mathbb{E}[w_q]\mathbb{E}[\mathcal{L}_p] + \mathrm{Cov}(w_q, \mathcal{L}_p) \big) - (|\mathcal{P}| - 1)\mathbb{E}[\mathcal{L}_p] \right) \\
&= \frac{1}{|\mathcal{P}|} \sum_{p \in \mathcal{P}} \left( \mathbb{E}[\mathcal{L}_p]\Big( \mathbb{E}[w_p] + (|\mathcal{P}| - 1)(1 - \sum_{q \neq p} \mathbb{E}[w_q]) \Big) + \mathrm{Cov}(w_p, \mathcal{L}_p) - (|\mathcal{P}| - 1) \sum_{q \neq p} \mathrm{Cov}(w_q, \mathcal{L}_p) \right)
\end{aligned}
$$

Since the routing weights are normalized, i.e., $\sum_{q \in \mathcal{P}} \mathbb{E}[w_q] = 1$, we have $1 - \sum_{q \neq p} \mathbb{E}[w_q] = \mathbb{E}[w_p]$. Substituting this identity simplifies the expectation-related term:

$$
\begin{aligned}
\mathrm{GE}(f) &\leq \frac{1}{|\mathcal{P}|} \sum_{p \in \mathcal{P}} \left( |\mathcal{P}| \, \mathbb{E}[\mathcal{L}_p]\mathbb{E}[w_p] + \mathrm{Cov}(w_p, \mathcal{L}_p) - (|\mathcal{P}| - 1) \sum_{q \neq p} \mathrm{Cov}(w_q, \mathcal{L}_p) \right) \\
&= \sum_{p \in \mathcal{P}} \left( \mathbb{E}[\mathcal{L}_p]\mathbb{E}[w_p] + \frac{1}{|\mathcal{P}|}\mathrm{Cov}(w_p, \mathcal{L}_p) - \frac{|\mathcal{P}| - 1}{|\mathcal{P}|} \sum_{q \neq p} \mathrm{Cov}(w_q, \mathcal{L}_p) \right). \tag{22} \\
&\leq \sum_{p \in \mathcal{P}} \left( \mathbb{E}[\mathcal{L}_p] + \frac{1}{|\mathcal{P}|}\mathrm{Cov}(w_p, \mathcal{L}_p) - \frac{|\mathcal{P}| - 1}{|\mathcal{P}|} \sum_{q \neq p} \mathrm{Cov}(w_q, \mathcal{L}_p) \right). \tag{23}
\end{aligned}
$$

Let $\hat{e}(\mathbf{f}_p)$ denote the empirical error of path $p$ on the training set. By standard Rademacher complexity arguments, with confidence $1 - \Delta$:

$$\mathbb{E}[\mathcal{L}_p] \leq \hat{e}(\mathbf{f}_p) + \mathcal{R}(|\mathcal{P}|, N, \Delta), \tag{24}$$

where $\mathcal{R}(|\mathcal{P}|, N, \Delta)$ denotes a Rademacher complexity–based term that depends on the number of paths $|\mathcal{P}|$, the training sample size $N$, and the confidence level $\Delta$.

Substituting into the previous decomposition, we obtain the final generalization bound:

$$\mathrm{GE}(f) \leq \sum_{p \in \mathcal{P}} \hat{e}(\mathbf{f}_p) + \sum_{p \in \mathcal{P}} \left[ \frac{1}{|\mathcal{P}|}\mathrm{Cov}(w_p, \mathcal{L}_p) - \frac{|\mathcal{P}| - 1}{|\mathcal{P}|} \sum_{q \neq p} \mathrm{Cov}(w_q, \mathcal{L}_p) \right] + \mathcal{R}(|\mathcal{P}|, N, \Delta). \tag{25}$$

$\square$

### B.2. Proof of Corollary 3.2

From Theorem 3.1, the generalization error bound contains the term

$$\sum_{p \in \mathcal{P}} \frac{1}{|\mathcal{P}|}\mathrm{Cov}(w_p, \mathcal{L}_p), \tag{26}$$

which is the contribution of each path's weight and its own loss. To minimize the upper bound, it is desirable that this covariance be negative:

$$\text{Cov}(w_p, \mathcal{L}_p) < 0, \quad \forall p \in \mathcal{P}. \tag{27}$$

For the cross-entropy loss, defined as $\mathcal{L}_p = -\sum_k y_k \log p_{k,p}$, where $y$ is a one-hot label vector, the loss reduces to $\mathcal{L}_p = -\log p_{\text{true},p}$. For the cross-entropy loss $\mathcal{L}_p = -\log p_{\text{true},p}$, the derivative with respect to $p_{\text{true},p}$ is strictly negative:

$$\frac{\partial \mathcal{L}_p}{\partial p_{\text{true},p}} = -\frac{1}{p_{\text{true},p}} < 0. \tag{28}$$

Thus, $\mathcal{L}_p$ is a monotonically decreasing function of $p_{\text{true},p}$.

By the property of covariance under a strictly monotone transformation, we have

$$\text{Cov}(w_p, \mathcal{L}_p) < 0 \quad \Longleftrightarrow \quad \text{Cov}(w_p, p_{\text{true},p}) > 0. \tag{29}$$

Therefore, to reduce the generalization error bound, a positive correlation between the path weight $w_p$ and the path's correctness probability $p_{\text{true},p}$ is required. □

### B.3. Proof of Corollary 3.3

Similarly, Theorem 3.1 includes the cross-path covariance term

$$-\sum_{p \in \mathcal{P}} \frac{|\mathcal{P}| - 1}{|\mathcal{P}|} \sum_{q \neq p} \text{Cov}(w_p, \mathcal{L}_q), \tag{30}$$

whose contribution to the bound is minimized if

$$\text{Cov}(w_p, \mathcal{L}_q) > 0, \quad \forall q \neq p. \tag{31}$$

Again, since $\mathcal{L}_q = -\log p_{\text{true},q}$ is monotonically decreasing in $p_{\text{true},q}$, the covariance sign flips under this transformation:

$$\text{Cov}(w_p, \mathcal{L}_q) > 0 \quad \Longleftrightarrow \quad \text{Cov}(w_p, p_{\text{true},q}) < 0. \tag{32}$$

This establishes that the weight of a given path should be negatively correlated with the correctness probabilities of all alternative paths. □

## C. Dataset

We evaluate the proposed C$^2$GMoE on six widely adopted graph benchmarks:

- **Amazon Photo & Computers:** Two co-purchase network datasets from Amazon. Nodes represent products, edges indicate frequent co-purchases. Node features are bag-of-words representations of product reviews, and labels correspond to product categories.

- **Coauthor CS:** A co-authorship network extracted from the Microsoft Academic Graph. Nodes represent authors, edges connect co-authors, and node features encode paper keywords. The task is to classify each author into a research field.

- **WikiCS:** A semi-supervised dataset derived from Wikipedia. Nodes represent Computer Science articles, edges represent hyperlinks. Node features are 300-dimensional GloVe word embeddings averaged over article content and references. The task is to classify articles into branches of computer science.

- **Flickr:** A social network dataset from the image-sharing platform. Nodes represent images. Edges are formed between images from the same location, submitted to the same gallery, group, or set, images sharing common tags, images taken by friends, etc. The task is to classify image categories based on visual content.

*Table 4.* Statistics of the datasets used in our experiments.

| Dataset | #Nodes | #Edges | #Features | #Classes | Split Ratio (Train/Val/Test) |
|---|---|---|---|---|---|
| Photo | 7,650 | 119,081 | 745 | 8 | 60%/20%/20% |
| Computers | 13,752 | 245,866 | 767 | 10 | 60%/20%/20% |
| CS | 18,333 | 81,894 | 6,805 | 15 | 60%/20%/20% |
| WikiCS | 11,701 | 216,123 | 300 | 10 | Official |
| Flickr | 89,250 | 899,756 | 500 | 7 | 50%/25%/25% |
| ogbn-arxiv | 169,343 | 1,166,243 | 128 | 40 | 53.7%/17.6%/28.7% |

- **ogbn-arxiv:** A large-scale citation network from the Open Graph Benchmark (OGB). Nodes correspond to Computer Science arXiv papers, directed edges denote citations. Each node is associated with a 128-dimensional Word2Vec embedding of paper titles and abstracts. The task is paper category prediction.

For WikiCS, Flickr and Ogbn-Arxiv, we follow the public split; for other datasets without public split, we generate random splits, distributing nodes into 60% training, 20% validation, and 20% testing partitions.

Table 4 summarizes the key characteristics of these datasets, including the number of nodes, edges, feature dimensions, and classes. We obtain the datasets from PyG (Fey & Lenssen, 2019).

## D. Baselines

We compare our proposed C$^2$GMoE framework against two categories of baselines covering representative GNN architectures and state-of-the-art MoE-based methods, with detailed descriptions as follows:

**1) Standard GNN Backbones:** These models represent canonical graph neural network architectures that differ in their aggregation and feature transformation mechanisms.

- **GCN** (Kipf & Welling, 2017) performs spectral-based graph convolution by aggregating features from immediate neighbors using a normalized adjacency matrix. Its propagation rule can be interpreted as a form of Laplacian smoothing, effectively blending neighborhood information in a linear fashion.

- **GraphSAGE** (Hamilton et al., 2017) introduces an inductive learning framework that samples fixed-size neighborhoods and aggregates features through learnable functions (e.g., mean, pooling). This design enables efficient training on large graphs and generalization to unseen nodes.

- **GIN** (Xu et al., 2019) is designed to emulate the expressive power of the Weisfeiler–Lehman graph isomorphism test. By using a learnable MLP after neighborhood summation, GIN achieves strong discriminative capability among graph structures.

**2) MoE-based Methods:** These methods adopt the Mixture of Experts (MoE) paradigm, which divides the model into multiple specialized expert networks and a gating mechanism to dynamically assign inputs to experts. We use the three standard GNNs above as expert backbones for fair comparison, ensuring consistency in the expert architecture across all MoE models.

- **GMoE** (Wang et al., 2023) introduces the Mixture-of-Experts (MoE) paradigm to graph neural networks to improve their adaptability to diverse training graph structures without incurring explosive computational overhead. The model allows individual nodes to dynamically select information aggregation experts. Different experts are trained to specialize in different structural patterns and incorporate information with varying hop sizes, where experts with larger hop sizes focus on aggregating longer-range information.

- **Mowst** (Zeng et al., 2024) proposes a mixture of weak and strong experts to decouple node self-feature modeling from neighborhood structure modeling. A lightweight MLP serves as the weak expert, while a standard GNN acts as the strong expert. A confidence mechanism based on the dispersion of the weak expert's prediction logits conditionally activates the strong expert, encouraging expert specialization while maintaining computational cost comparable to a single GNN.

- **DA-MoE** (Yao et al., 2025) addresses the depth-sensitivity issue in graph neural networks, where the optimal number of GNN layers varies with graph scale. It treats GNNs of different depths as distinct experts, enabling information aggregation at multiple scales. In addition, DA-MoE employs a GNN-based gating network to incorporate structural information into expert selection, allowing adaptive assignment of graph instances to experts with appropriate depths.

## E. Computational Complexity Analysis

We evaluate complexity via FLOPs to demonstrate that C²GMoE maintains efficiency comparable to standard GNNs. For a standard GNN layer with hidden dimension $d_0$, the complexity is approximately $\mathcal{O}(|E|d_0 + Nd_0^2)$, where the feature transformation $Nd_0^2$ typically dominates the overhead. For a layer with $N$ nodes, $|E|$ edges, and $M$ total experts (where $1 + k$ are active), we control the cost by scaling the hidden dimension $d = d_0/\sqrt{1+k}$ relative to a standard GNN's dimension $d_0$.

Expert Computation: The feature transformation cost, which typically dominates GNN overhead, becomes $(1 + k) \cdot N(d_0/\sqrt{1+k})^2 = Nd_0^2$, aligning with standard GNNs. The neighbor aggregation cost becomes $\sqrt{1+k}|E|d_0$, which grows only sub-linearly and remains efficient for small $k$.

Auxiliary Overhead: Additional modules such as the gating network $\mathcal{O}(NdM)$, contrastive routing $\mathcal{O}(Ngd)$, and confidence fusion $\mathcal{O}(Nd)$ all possess linear complexity w.r.t. $N$. These node-level operations incur FLOPs significantly lower than the core message-passing bottleneck.

During inference, the contrastive routing module is discarded, ensuring the total FLOPs remain on the same order of magnitude as standard GNNs while providing significantly higher model capacity.

## F. Experiments Details

### F.1. Software and Hardware

All experiments are implemented in Python 3.10.12 with PyTorch 2.0.1 and PyTorch Geometric (PyG) 2.6.1, using CUDA 11.7. All computations are conducted on a single server equipped with an NVIDIA V100 GPU with 32 GB of memory.

### F.2. Implementation Details.

For Flickr and ogbn-arxiv, the number of layers ($L$) is fixed to 3 and the hidden dimension ($d$) to 256, following the original papers (Zeng et al., 2020; Hu et al., 2020). For other datasets, we fix the number of layers to 2 and select the hidden dimension by grid search on vanilla GNN backbones. To maintain comparable FLOPs, the hidden dimension of C²GMoE is adjusted based on routing sparsity $k$. For other MoE-based baselines, we follow the architectural choices and hyperparameter settings reported in their original papers.

### F.3. Hyperparameter Settings

We implement our C²GMoE framework using a fixed number of experts $M = 8$ across all datasets. For fair comparisons, we set the number of layers be the same as that of the corresponding vanilla GNN, and the hidden dimension is adjusted according to the number of active experts $k$. We employ the Adam optimizer (Kingma & Ba, 2015) to train the model, and the optimal hyperparameters are determined via grid search within the following search space:

- Learning rate within $\{1e-2, 1e-3\}$.

- Number of active experts $k$ within $\{2, 3, 4\}$.

- Weight decay within $\{0, 1e-5, 5e-5, 1e-4, 5e-4\}$.

- Dropout ratio within $\{0.1, 0.2, 0.3, 0.4, 0.5\}$.

For the trade-off parameters in our objective function, we fix both the routing contrastive weight $\alpha$ and the confidence fusion weight $\gamma$ at 0.1 to maintain balanced supervision during training. Target number of groups for clustering: 20. Layer Normalization is applied to the output of each expert to stabilize the distribution of hidden representations. We adopt a full-batch training strategy for all datasets.

### F.4. Baselines

We implement GCN, SAGE, GIN using PyG (Fey & Lenssen, 2019). For GCN, SAGE, and GIN, we conduct a grid search over the following hyperparameter space:

- Learning rate: $\{0.1, 0.01, 0.001\}$

- Dropout ratio: $\{0.1, 0.2, 0.3, 0.4, 0.5\}$

- Hidden feature dimension: $\{64, 128, 256, 512\}$

For all other baselines, we use the official implementations and adopt the hyperparameter search spaces described in the original papers. The sources are listed as follows:

- GMoE: https://github.com/VITA-Group/Graph-Mixture-of-Experts

- Mowst: https://github.com/facebookresearch/mowst-gnn

- DA-MoE: https://github.com/Celin-Yao/DA-MoE

Specifically, for Mowst, the hyperparameters are tuned as follows:

- Learning rate: $\{0.1, 0.01, 0.001\}$

- Dropout ratio: $\{0.1, 0.2, 0.3, 0.4, 0.5\}$

- Gating network learning rate: $\{0.1, 0.01, 0.001\}$

- Gating network dropout ratio: $\{0.1, 0.2, 0.3, 0.4, 0.5\}$

For GMoE, the hyperparameters are tuned as follows:

- Total number of experts $n$: $\{4, 8\}$

- Number of hop-1 experts $m$: $\{0, n/2, n\}$

- Number of active experts $k$: $\{1, 2, 4\}$

- Loss trade-off weight $\lambda$: $\{0.1, 1\}$

The hyperparameter configuration achieving the best performance on the validation set is used to report results on the test set.

## G. More Experimental Results

### G.1. Performance Evaluation with GIN Experts

Table 5 reports the classification performance with GIN experts, which demenstrates that our performance gains are robust against variations in expert backbone architectures.

### G.2. Hyperparameter Sensitivity Analysis

We conduct additional sensitivity experiments to evaluate the robustness of C$^2$GMoE. As presented in Table 6 and Table 7, the model performance fluctuates by less than 0.5% across a wide spectrum of configurations for the loss weights $\alpha$ and $\gamma$ within $[0.001, 2.0]$, with a slight degradation observed only when weights are excessively large. Similarly, the framework exhibits high stability across different temperature coefficients $\tau$ (Table 8), where $\tau = 0.2$ consistently yields optimal performance on both the Photo and WikiCS datasets.

*Table 5.* Performance comparison of different methods with GIN experts (%).

| Dataset | Metric | GIN | GMoE-GIN | Mowst-GIN | DAMoE-GIN | C$^2$GMoE-GIN |
|---------|--------|-----|----------|-----------|-----------|----------------|
| Photo | Micro-F1 | $\underline{94.31 \pm 0.20}$ | $93.91 \pm 0.50$ | $93.52 \pm 1.12$ | $92.38 \pm 0.93$ | $\mathbf{94.82 \pm 0.25}$ |
|  | Macro-F1 | $\underline{93.82 \pm 0.28}$ | $93.20 \pm 0.59$ | $93.12 \pm 1.09$ | $92.10 \pm 0.99$ | $\mathbf{93.97 \pm 0.30}$ |
| WikiCS | Micro-F1 | $76.68 \pm 0.99$ | $\underline{78.49 \pm 0.52}$ | $75.08 \pm 0.95$ | $76.90 \pm 1.20$ | $\mathbf{79.30 \pm 0.61}$ |
|  | Macro-F1 | $73.53 \pm 1.39$ | $\underline{74.82 \pm 0.85}$ | $71.62 \pm 1.24$ | $73.10 \pm 1.67$ | $\mathbf{76.06 \pm 0.88}$ |
| Computers | Micro-F1 | $90.69 \pm 0.51$ | $91.49 \pm 0.38$ | $\underline{91.93 \pm 0.11}$ | $89.40 \pm 1.06$ | $\mathbf{92.47 \pm 0.15}$ |
|  | Macro-F1 | $89.70 \pm 0.45$ | $90.79 \pm 0.63$ | $\underline{90.85 \pm 0.16}$ | $87.09 \pm 1.42$ | $\mathbf{91.77 \pm 0.11}$ |
| CS | Micro-F1 | $92.71 \pm 0.04$ | $93.47 \pm 0.59$ | $\underline{93.60 \pm 0.16}$ | $92.88 \pm 0.16$ | $\mathbf{93.95 \pm 0.13}$ |
|  | Macro-F1 | $91.07 \pm 0.02$ | $91.81 \pm 0.79$ | $\underline{91.85 \pm 0.29}$ | $90.68 \pm 0.25$ | $\mathbf{92.18 \pm 0.16}$ |
| Flickr | Micro-F1 | $53.81 \pm 0.32$ | $54.70 \pm 0.22$ | $\underline{54.72 \pm 0.47}$ | $53.91 \pm 0.23$ | $\mathbf{55.59 \pm 0.19}$ |
|  | Macro-F1 | $22.58 \pm 0.75$ | $24.50 \pm 0.74$ | $\underline{24.88 \pm 0.76}$ | $24.07 \pm 0.95$ | $\mathbf{25.68 \pm 0.45}$ |
| ogbn-arxiv | Micro-F1 | $70.37 \pm 0.22$ | $71.25 \pm 0.25$ | $\underline{71.36 \pm 0.14}$ | $70.89 \pm 0.26$ | $\mathbf{71.68 \pm 0.23}$ |
|  | Macro-F1 | $47.73 \pm 0.41$ | $47.50 \pm 1.29$ | $\underline{51.09 \pm 0.21}$ | $50.25 \pm 0.75$ | $\mathbf{51.24 \pm 0.16}$ |

*Table 6.* The parameter sensitivity of C$^2$GMoE for different $\alpha$ on Photo and WikiCS datasets.

| Dataset | 0.001 | 0.01 | 0.1 | 0.5 | 1.0 | 2.0 |
|---------|-------|------|-----|-----|-----|-----|
| Photo | 95.07 | 95.10 | 95.27 | 95.03 | 94.90 | 94.85 |
| WikiCS | 80.22 | 80.21 | 80.11 | 79.90 | 79.46 | 79.86 |

*Table 7.* The parameter sensitivity of C$^2$GMoE for different $\gamma$ on Photo and WikiCS datasets.

| Dataset | 0.001 | 0.01 | 0.1 | 0.5 | 1.0 | 2.0 |
|---------|-------|------|-----|-----|-----|-----|
| Photo | 95.15 | 95.29 | 95.27 | 95.23 | 95.19 | 94.95 |
| WikiCS | 80.01 | 80.00 | 80.11 | 80.25 | 80.17 | 79.98 |

*Table 8.* The parameter sensitivity of C$^2$GMoE for different $\tau$ on Photo and WikiCS datasets.

| Dataset | 0.05 | 0.1 | 0.2 | 0.5 | 1.0 |
|---------|------|-----|-----|-----|-----|
| Photo | 94.92 | 95.16 | 95.27 | 94.98 | 95.01 |
| WikiCS | 79.96 | 80.09 | 80.11 | 79.99 | 80.06 |

