# OpenReview forum: "Discriminative Mixture-of-Experts on Graphs with Reliable Expert Fusion"
_ICML.cc/2026/Conference — ICML 2026 regular_

### Official Review · Reviewer_pfPc · 2026-03-09

**Soundness:** 3
**Presentation:** 4
**Significance:** 3
**Originality:** 3
**Overall Recommendation:** 5
**Confidence:** 4

**Summary:**

This paper studies the problem of graph MOE. Specifically, this paper discovers two critical phenomena through extensive studies, i.e., discirmination loss and routing uncertainty, revealing that the current graph MOE methods fail to fully realize the architectural potential. These findings further motive to propose a novel graph MOE method with theoretical analysis that integrates group-wise constrative routing restore to discriminative power by promoting expert specialization under load-balancing constraints. Extensive experiments well demonstrate the effectiveness of the proposed method.

**Compliance With Llm Reviewing Policy:**

Affirmed.

**Key Questions For Authors:**

Please refer to the weakness section.

**Limitations:**

No. I didn’t find there is potential negative societal impact of their work. The authors may analyze the complexity of the method on large graphs.

**Strengths And Weaknesses:**

Strengths:
- The paper is well motivated. I think most important claims of the paper are well justified. Specifically, the authors point out that there are two fundamental problems when designing graph MOE using extensive empirical studies, where I think this problems are worth investigating, and each of these problems is addressed by proposing specific designs in the proposed method. Also, the proposed method is well supported by theoretical analysis, and the experiments are well designed.
- The presentation of the paper is consistent with ICML standards. Overall, the paper is well written and structured, which is easy to follow.
- The paper provides new understanding on the existing methods, e.g., the discrimination loss and routing uncertainty, and the theoretical analysis can help better understand the property of the proposed method. The paper did a good job on advancing the graph MOE.

Weaknesses:

- The method uses the logits to represent the confidence, so what the is the relationship between the logits and the confidence? More explanations are needed.
- In fig.3, the results show that a vast majority of nodes exhibit high routing uncertainty, which are quite interesting. Could the authors provide some explanations on the results?
- There are two parameters in the overall loss function, and I suggest the authors provide the performance analysis on the two parameters.
- Considering that most of the current graph learning methods are combined with the LLM, it is beneficial that the authors discuss the advantages and disadvantages of graph MOE compared with them.

---

> ### Author Rebuttal · Authors · 2026-03-29
>
> We thank the reviewer for the positive feedback and insightful comments. We address the specific points below:
>
> **W1: Relationship between Logits and Confidence**
>
> We clarify that in our framework, confidence is not represented by raw logit values. Instead, the expert output logits are first transformed into class probabilities via Softmax. During training, the probability corresponding to the ground-truth label is defined as the target confidence. We then train a confidence predictor to estimate this true predictive correctness. During inference, the predictor outputs this estimated confidence to guide fusion. This approach ensures that our confidence reflects the actual reliability of an expert’s prediction, rather than being a mere reflection of logit magnitude or dispersion.
>
> **W2: Explaining the Prevalent Routing Uncertainty in Figure 3**
>
> The routing module in GMoE assigns weights based on node embeddings but lacks explicit supervision to differentiate experts. As a result, the router tends to adopt an averaging strategy, which reduces per-sample risk but suppresses expert specialization. Furthermore, different experts often learn highly similar representations, making it difficult for the router to distinguish which expert is more suitable for a given node. This combination leads to widespread high routing uncertainty.
>
> **W3: Hyperparameter Analysis in the Loss Function**
>
> We thank the reviewer for this suggestion and have conducted additional experiments. For the loss weights $\lambda_1$ and $\lambda_2$, the performance fluctuates by less than 0.5% across a wide range ([0.001, 2.0]), with a slight decrease observed only when weights are excessively large.
>
> Sensitivity to $\lambda_1$:
>
> | dataset | 0.001 | 0.01 | 0.1 | 0.5 | 1   | 2   |
> | --- | --- | --- | --- | --- | --- | --- |
> | photo | 95.07 | 95.10 | 95.27 | 95.03 | 94.90 | 94.85 |
> | wikics | 80.22 | 80.21 | 80.11 | 79.90 | 79.46 | 79.86 |
>
> Sensitivity to $\lambda_2$:
>
> | dataset | 0.001 | 0.01 | 0.1 | 0.5 | 1   | 2   |
> | --- | --- | --- | --- | --- | --- | --- |
> | photo | 95.15 | 95.29 | 95.27 | 95.23 | 95.19 | 94.95 |
> | wikics | 80.01 | 80.00 | 80.11 | 80.25 | 80.17 | 79.98 |
>
> **W4: Comparison between Graph-MoE and Graph-LLM methods**
>
> This is a forward-looking perspective. Graph-MoE excels in specialized structural modeling by allocating experts to capture diverse local patterns, leading to finer-grained topology capture and significantly higher computational efficiency. However, it lacks the deep semantic understanding and text-reasoning capabilities of LLMs. In contrast, Graph-LLMs possess superior zero-shot transfer and cross-domain generalization, especially on text-rich graphs, but they struggle to directly process complex structural information and incur higher computational costs.
>
> **Limitations**: We have included a computational complexity analysis. The results show that the total FLOPs remain on the same order of magnitude as standard GNNs. Please refer to our response to Reviewer [YZwz] (W4: Computational Complexity) for a detailed analysis. We will add a Limitations section in the final version to analyze the complexity of the method on large graphs.

---

> > ### Author Rebuttal · Reviewer_pfPc · 2026-04-05
> >
> > Thank you. My questions have been adequately answered.

---

### Official Review · Reviewer_U4XJ · 2026-03-12

**Soundness:** 3
**Presentation:** 3
**Significance:** 2
**Originality:** 1
**Overall Recommendation:** 3
**Confidence:** 4

**Summary:**

This paper proposes C^2GMoE, a Graph Mixture of Experts (MoE) framework with Contrastive Routing Learning and Confidence-aware Fusion. C^2GMoE is motivated by empirical results showing that an existing graph MoE (GMoE) suffers from expert homogenization, routing collapse, and routing uncertainty on three benchmarks. To address these issues, C^2GMoE combines group-wise contrastive routing and confidence-aware fusion. More specifically, nodes are grouped based on attributes and structure and a routing contrastive loss induces the routing for each node to be similar to its group and different from other groups. Moreover, motivated by a theoretical result, expert predictions are aggregated based on weights that combine individual and relative confidence scores.  In the experiments, C^2GMoE is compared against three Graph MoE baselines using six datasets in terms of node classification. The results show that the proposed model outperforms the baselines. Additional ablation studies and analysis show the contribution of the group-wise contrastive routing and the confidence-aware fusion Moreover, C^2GMoE better distributes the load among experts and leads to lower homogenization.

**Compliance With Llm Reviewing Policy:**

Affirmed.

**Final Justification:**

I've checked the reviews and rebuttal and decided to keep my original recommendation (Weak Reject). My review is the most negative one but the rebuttal did not fully address my concerns. In particular, all optimizations proposed by previous work to address the same issues identified in the paper can in principle be adapted to the graph case. Simply stating that the paper focuses on graph data is not sufficient as it often requires only the use of an appropriate encoder for the data. The proposed solution also achieves small gains compared to the baselines without fully discussing the tradeoff between the gains and the additional training and/or inference time.

**Key Questions For Authors:**

- Q1. What is specific about graphs regarding the issues identified in Section 2?

- Q2. Why existing solutions that attempt to address the issues identified can not be easily translated to graphs?

- Q3. How do the contributions of this paper overlap with those form Cao et al. (2024)?

**Limitations:**

The paper does not discuss its limitations explicitly.

**Strengths And Weaknesses:**

**Strengths:**
- S.1. The paper is clear and easy to follow
- S.2. The proposed weighting mechanism is supported by theory
- S.3. The problems addressed in the paper are motivated by empirical results

**Weaknesses:**
- W.1. The issues identified in the paper do not seem to be particularly about graphs
- W.2. Theorem 3.1. seems to be borrowed from the literature but is proved again
- W.3. Some of the gains reported in Tables 1-3 seem minor

**Detailed comments:**

This paper addresses relevant problems in MoEs in the context of graphs and proposes a solution that seems to have some novel elements. However, my main concern is how the issues identified differ in the context of graphs vs. in other MoE applications. See a description of the weaknesses below:

**MoEs for graphs vs. other types of data:**

The empirical limitations identified in Section 2 (expert homogenization, routing collapse, and routing uncertainty) do not seem to affect only graph data. Previous work has highlighted and attempted to address them for other data domains. It is unclear whether these limitations need a solution that is specific for graphs or if existing solutions can be borrowed. Some examples:

@article{wen2025adaptive,
  title={Adaptive-expert-weight-based load balance scheme for dynamic routing of MoE},
  author={Wen, Jialin and Li, Xiaojun and Yao, Junping and Kong, Xinyan and Cheng, Peng},
  journal={Frontiers in Neurorobotics},
  volume={19},
  pages={1590994},
  year={2025},
  publisher={Frontiers Media SA}
}

@inproceedings{dai2022stablemoe,
  title={Stablemoe: Stable routing strategy for mixture of experts},
  author={Dai, Damai and Dong, Li and Ma, Shuming and Zheng, Bo and Sui, Zhifang and Chang, Baobao and Wei, Furu},
  booktitle={Proceedings of the 60th Annual Meeting of the Association for Computational Linguistics (Volume 1: Long Papers)},
  pages={7085--7095},
  year={2022}
}

@article{wu2024gw,
  title={Gw-moe: Resolving uncertainty in moe router with global workspace theory},
  author={Wu, Haoze and Qiu, Zihan and Wang, Zili and Zhao, Hang and Fu, Jie},
  journal={arXiv preprint arXiv:2406.12375},
  year={2024}
}

@article{dialameh2025bayesian,
  title={Bayesian Mixture of Experts For Large Language Models},
  author={Dialameh, Maryam and Rajabzadeh, Hossein and Zhang, Weiwei and Ahmed, Walid and Kwon, Hyock Ju},
  journal={arXiv preprint arXiv:2511.08968},
  year={2025}
}


**Theorem 3.1:**

Theorem 3.1 and Section 3.3. more broadly seem to rely heavily on the results from Cao et al. (2024). While the paper uses the word “revisit” before describing the result, I still believe that the reader might assume that the result is at least partially novel. The paper requires a more precise and detailed comparison between the contributions from Cao et al. (2024) and the new contributions claimed.

**Performance gains:**

The performance gains presented in Tables 1-3 seem minor. Some of them might need a statistical test to verify significance. It is hard to argue for a more complex method that achieves small gains. In particular, the computational cost of C^2GMoE should also be compared with the baselines. In case the costs are significantly higher, it is possible that many users would simply opt for the baseline.

---

> ### Author Rebuttal · Authors · 2026-03-29
>
> We thank the reviewer for the detailed feedback. We address your concerns below:
>
> **Q1&W1: Graph-specific Challenges**
>
> While previous studies on phenomena like routing collapse have provided general insights, their existence and impact within graph settings remains unclear. We are the first to systematically verify these phenomena in graphs. On graphs, our empirical evidence reveals that such issues are amplified and driven by distinct underlying mechanisms unique to graph data. Unlike i.i.d. data, nodes in a graph are interdependent, with their semantics jointly determined by both individual features and topology. Consequently, existing strategies inherited from NLP lack an awareness of structural consistency, rendering them unable to capture shared expert patterns when applied to graphs. The identified challenges are not merely inherited from general MoE but are fundamentally shaped by graph structure.
>
> **Q2: Limitations of Translating Existing MoE Solutions to Graphs.**
>
> Previous works do not focus on graph domain and fail to account for its inherent structural dependencies. Our method integrates feature and structural semantics, enabling graph-tailored expert specialization, which cannot be achieved by existing solutions.
>
> Regarding the reviewer's mentioned methods:
>
> - Wen et al. (2025) propose a load balancing strategy. Without considering semantic or structural information, their method may cause suboptimal assignments in graphs. For example, if an expert specializes in a frequent, overloaded subgraph, the method suppresses its weight, forcing structurally similar nodes to less suitable experts.
>
> - GW-MoE addresses routing uncertainty by broadcasting uncertain tokens to all experts during fine-tuning. This fails to eliminate uncertainty and leads to redundant expert usage. Conversely, we model confidence to learn reliable fusion weights and directly address routing uncertainty.
>
> - StableMoE and Bayesian-MoE focus on different problems from ours, which leads to very different methods. StableMoE focuses on routing instability, while our work focuses on expert specialization and routing uncertainty. Bayesian-MoE targets posterior calibration, fundamentally differing from our goal of resolving routing uncertainty in Graph-MoE.
>
>
> **Q3: Overlap with Cao et al. (2024)**
>
> The overlap is limited to the general theoretical framework, while our core contributions differ in two dimensions:
>
> - Distint Domains and Challenges：Cao et al. focus on multimodal fusion in open environments, addressing modality imbalance, and external noise. Our work is dedicated to MoE on graphs, targets two bottlenecks on Graph-MoE: discriminative loss and routing uncertainty.
>
> - Unique Model and Derivations：Cao et al. operate on modality-level fusion in a single step. In contrast, our model performs expert-level aggregation within a multi-layer routing GNN. In this architecture, routing decisions affect subsequent layers and uncertainty accumulates across layers. This shifts the analysis from a single-step setting to a multi-layer, path-dependent setting, requiring different adaptations in the derivation process. In the Graph-MoE, we further incorporate confidence fusion at the decision layer to resolve routing uncertainty while maintaining efficiency.
>
>
> We will clarify in the final version.
>
> **W2：About Theorem 3.1**
>
> Theorem 3.1 is not a "borrowing" from existing literature, nor is it a re-proof. Firstly, it is specifically adapted and instantiated within the context of multi-layer Graph-MoE and translated into an effective model design to handle the routing uncertainty. Secondly, our derivation differs due to the different model architecture. In our setting, each layer is an MoE layer in a multi-layer GNN, meaning that routing decisions affect subsequent layers and uncertainty accumulates across layers. This shifts the analysis from a single-step setting to a multi-layer, path-dependent setting, requiring corresponding adaptations in derivation, including the derivation process and formulation of weights.
>
> **W3: Performance Gains and Computational Cost**
>
> We conducted paired t-tests over 10 runs. **The improvements are statistically significant across multiple datasets and backbones.**
>
> | Dataset | Backbone | Second-best | C²GMoE | p-value |
> | --- | --- | --- | --- | --- |
> | Photo | gcn | 94.55 ± 0.30 | 95.16 ± 0.20 | < 0.001 |
> | Wikics | gcn | 79.73 ± 0.73 | 80.11 ± 0.63 | < 0.05 |
> | Computers | gcn | 91.75 ± 0.48 | 92.88 ± 0.18 | < 0.001 |
> | Photo | sage | 95.71 ± 0.19 | 96.19 ± 0.14 | < 0.001 |
> | Wikics | sage | 79.14 ± 0.55 | 80.09 ± 0.53 | < 0.01 |
> | Computers | sage | 91.75 ± 0.48 | 92.37 ± 0.18 | < 0.01 |
>
> For table 3, paired t-tests also show that full model performs significantly better than the ablated variants (p < 0.05).
>
> Due to space constraints, please refer to our response to Reviewer YZwz (W4: Computational Complexity) for cost analysis.
>
> Limitations: We will add a Limitations section in the final version.

---

> > ### Author Rebuttal · Reviewer_U4XJ · 2026-04-04
> >
> > The rebuttal only partially addressed my concerns. The statement of the theorem and the proof strategy are too similar to the results from Cao et al. (2024), which means that the differences stated do not seem to impact the final statement of the theorem. The comments regarding some of the previous work are subjective. And the gains are still small (although significant based on the response). I believe that weak reject still reflects my recommendation regarding this paper.

---

> > > ### Author Response · Authors · 2026-04-07
> > >
> > > > About the theorem
> > >
> > > The similarity in the theorem's form is due to the use of a consistent theoretical analytical framework and common Rademacher complexity foundation. However, this framework has been adapted to different problems and models, where the internal meanings of variables, application scenarios, and the guidance provided for model design have changed. The novelty does not lie in a new theoretical framework, but in its adaptation to the Graph-MoE setting and the resulting confidence-aware fusion mechanism to address the routing uncertainty we identified. Specifically:
> > >
> > > - Different Domains and Challenges: Cao et al. focus on multimodal fusion in open environments, addressing issues like modality imbalance and external noise. In contrast, our work is dedicated to MoE on graphs, targeting two specific bottlenecks in Graph-MoE: discrimination loss and routing uncertainty. These two issues are systematically revealed for the first time in this paper.
> > >
> > > - Different Model Architectures: Cao et al. focus on single-layer modality fusion, where each modality generates an independent prediction followed by a weighted aggregation. In contrast, our model is a multi-layer MoE GNN, where each node's representation is refined through a stack of expert layers. Specifically, the expert selection at each layer is conditioned on the output of the preceding layer, causing routing uncertainty to accumulate across the hierarchy. Such an architecture necessitates handling path-dependent expert compositions rather than single-step fusion.
> > >
> > > - Different Derivations: In contrast to the single-step analysis in Cao et al., our derivation must recursively handle expert selection across multiple layers, requiring different adaptations in the derivation process (e.g., the definitions of weights and losses, and the specific application of Jensen's inequality).
> > >
> > >
> > > > The comments regarding some of the previous work are subjective
> > >
> > > While the reviewer mentions that our comments are subjective, it is not specified which particular analysis regarding the limitations of prior works is inaccurate. We welcome further discussion on any specific technical details of these prior works to ensure our comparison remains as precise as possible.
> > >
> > > To clarify, we provide a more detailed analysis. Existing MoE methods effectively solve the problems they target in their original settings, but they are not designed for graph-structured data. As a result, their modeling objectives do not fully align with the challenges we address. This mismatch stems from differences in problem settings and design goals, rather than subjective preference.
> > >
> > > - While Wen et al. (2025) effectively addresses load balancing by adjusting expert weights based on historical activation values, their approach primarily focuses on efficient use of computational resources. In contrast, our primary objective is to resolve Discrimination Loss—ensuring that different GNN experts capture diverse, non-redundant graph patterns while the router precisely aligns node structural roles with expert specializations. Our goal therefore goes beyond load balancing, and directly applying such methods does not fully align with the problem we study. Such methods can only guarantee load balancing, but may undermine structure-consistent expert specialization in Graph-MoE.
> > >
> > > - GW-MoE manages routing uncertainty by broadcasting uncertain tokens to all experts during fine-tuning. While this mitigates the impact of incorrect routing, it does not resolve the root cause of uncertainty. In graph settings, routing uncertainty is prevalent across a majority of nodes (as shown in our Figure 3). If such a large proportion of nodes were broadcasted to all experts, the MoE would effectively degenerate into a dense model, violating the fundamental design principle of sparse activation and expert labor division. Our objective is to learn reliable fusion weights via confidence modeling to resolve this uncertainty without sacrificing sparsity.
> > >
> > > - Bayesian-MoE and StableMoE address different problems than our work, which naturally results in distinct technical approaches. Specifically, Bayesian-MoE aims at posterior calibration, which fundamentally differs from our objective of resolving routing uncertainty in Graph-MoE. StableMoE concentrates on routing instability, whereas our work targets expert specialization and routing uncertainty.
> > >
> > >
> > > We will incorporate these distinctions and formal citations into the final revision.
> > >
> > > > About the gains
> > >
> > > We have conducted 10 independent runs with paired t-tests, which confirm that these gains are statistically significant across multiple datasets and backbones (as shown in the paired t-test results table provided in our initial response). These improvements are obtained on highly competitive benchmarks, which is meaningful as it signifies advancements in more challenging cases such as long-tail nodes and uncertain nodes.

---

### Official Review · Reviewer_Tndk · 2026-03-13

**Soundness:** 3
**Presentation:** 4
**Significance:** 3
**Originality:** 4
**Overall Recommendation:** 4
**Confidence:** 4

**Summary:**

This article's central area comprises the architectural refinement and optimization of Graph Mixture-of-Experts (Graph-MoE) models. The submission proceeds to assess a notable context: the dual bottlenecks of "Discrimination Loss" (where GNN experts homogenize into overlapping representation spaces and routers collapse) and "Routing Uncertainty" (where high-entropy expert assignments actively degrade model performance). To mitigate these underlying flaws, the authors propose C²GMoE, a framework introducing a group-wise Contrastive routing strategy to enforce expert specialization via semantic clustering, alongside a Confidence-aware fusion mechanism grounded in a theoretical generalization error bound. Extensive experiments across six benchmarks indicate consistent improvements over existing Graph-MoE baselines.

**Compliance With Llm Reviewing Policy:**

Affirmed.

**Final Justification:**

The rebuttal meaningfully clarified the issues I raised. I remain my original assessment.

**Key Questions For Authors:**

1. Computational Complexity: Can you provide a rigorous analysis of the time and memory complexity of C²GMoE compared to the baselines? Specifically, what is the empirical overhead (in milliseconds/epochs and GPU memory) of the Confidence Predictor and the routing contrastive loss?
2. Theoretical Gap: How do you justify the assumption of linear or convex transformations in Theorem 3.1 when utilizing highly non-linear GNN backbones? Does the empirical performance hold up because of, or in spite of, this theoretical bound?
3. Scalability: The current implementation uses full-batch training. How compatible is C²GMoE with standard scalable graph sampling techniques (e.g., GraphSAINT, NeighborSampler)? Would the contrastive routing strategy collapse if only operating on mini-batch subgraphs?
4. Hyperparameter Sensitivity: How sensitive is the model to the target number of groups $C$ (currently set to 20) and the temperature parameter $\tau$?

**Limitations:**

No. The paper should include a limitations section discussing: computational complexity and efficiency concerns、applicability to dynamic or temporal graphs and performance on extremely large-scale graphs requiring sampling.

**Strengths And Weaknesses:**

Strengths:

Compelling Empirical Motivation:The paper excels in its problem identification. Revealing that existing Graph-MoE methods suffer from severe expert homogenization (demonstrated by CKA similarities frequently exceeding 0.8) and pervasive routing collapse provides a strong, concrete motivation for the proposed architecture.

Theoretical Grounding for Architectural Choices:The derivation of the generalization error bound (Theorem 3.1) establishes a principled foundation for the confidence-aware fusion mechanism, moving beyond heuristic gating to a mathematically justified weight design.

Comprehensive Empirical Validation: The methodology is rigorously evaluated across 6 diverse benchmarks using multiple distinct GNN backbones (GCN, GraphSAGE, GIN), proving that the C²GMoE framework is model-agnostic and consistently outperforms recent baselines like GMoE, Mowst, and DA-MoE.

Weaknesses:

Theoretical Over-simplification:The mathematical rigor in Theorem 3.1 is undermined by a remarkably strong assumption: treating expert functions as linear or non-decreasing convex transformations. Modern GNNs rely heavily on non-linear activations (e.g., ReLU, GELU), making this assumption unrealistic and potentially invalidating the theoretical guarantees in practical deployment.

Scalability and Computational Complexity:The paper relies on full-batch training, which is misaligned with the current trajectory of large-scale graph learning. There is a glaring absence of formal time and space complexity analysis ($\mathcal{O}(\dots)$). The computational overhead of maintaining multiple experts, computing contrastive losses, and estimating confidence scores makes the practical utility of this method on web-scale graphs questionable.

Hyperparameter Brittleness: While the ablation on active experts ($k$) and total experts ($M$) is appreciated, the framework introduces a highly complex hyperparameter space (e.g., loss trade-off weights $\alpha$ and $\gamma$, temperature $\tau$, and the number of semantic groups $C$). The lack of sensitivity analysis for these variables leaves the robustness of the model ambiguous.

---

> ### Author Rebuttal · Authors · 2026-03-29
>
> We thank the reviewer for the detailed and constructive comments. We address the concerns and questions below:
>
> **Q1&W2: Computational Complexity**
>
> 1. Time complexity：
>
>   Due to space constraints, please refer to our response to Reviewer [YZwz] (W4: Computational Complexity) for a more detailed derivation. The results show that the total FLOPs remain on the same order of magnitude as standard GNNs.
>
> 2. Space complexity:
>
>   The spatial complexity primarily stems from model parameters and intermediate representations. Regarding parameter storage, the complexity of the expert networks is $O(L \cdot (M+1) \cdot P_{0})$, the gating network is $O(d \cdot M)$, the contrastive routing prototypes are $O(L \cdot C \cdot M)$, and the confidence predictor in the final layer is $O(d \cdot M)$. Regarding intermediate representations, the scale of node representations that need to be stored is $O(L \cdot N \cdot d)$.
>
>   Overall, the space complexity is dominated by expert parameters and node representations, resulting in $O(L \cdot M \cdot P_{0} + L \cdot N \cdot d)$. It grows linearly with $M$ and has a larger parameter size than a single GNN, yet achieves capacity expansion through sparse activation without compromising inference efficiency.
>
>
> We report the additional computational cost (in milliseconds per epoch) and GPU memory consumption (in MB) incurred by the confidence predictor and the routing contrastive loss on the Photo dataset in the table below. The overhead introduced by these modules is minimal.
>
> | Module | Time Cost (ms/epoch) | Memory Usage (MB) |
> | --- | --- | --- |
> | Routing Contrastive Loss | 73.73 | 60.88 |
> | Confidence Predictor | 0.64 | 1.43 |
>
> **Q2&W1: Justification of Theoretical Assumptions in Theorem 3.1**
>
> The assumption of linear or convex transformations is a common and necessary simplification in deep learning theory to ensure mathematical tractability[1-3]. In practice, most widely used backbones (e.g., GCN, GIN) utilize ReLU activations, making them locally piecewise linear. As a result, the model behaves approximately as a linear operator within each local activation region. This property allows us to approximate expert transformations as linear mappings in our theoretical analysis, enabling tractable and interpretable results.
>
> **Q3&W2: Scalability and Compatibility with Mini-batch Sampling**
>
> We would like to clarify that C²GMoE is fully compatible with mini-batch training. The contrastive routing loss is defined at the node level and does not require full-graph information. Under standard sampling strategies, contrastive learning can be performed within sampled subgraphs without collapse. In practice, semantic groups can be precomputed globally to maintain consistency, while group prototypes are computed dynamically within each batch. This design preserves both efficiency and stability, and does not rely on full-batch training.
>
> **Q4&W3: Hyperparameter Sensitivity**
>
> We thank the reviewer for this suggestion and have conducted additional experiments. For the loss weights $\lambda_1$ and $\lambda_2$, the performance fluctuates by less than 0.5% across a wide range ([0.001, 2.0]), with a slight decrease observed only when weights are excessively large. Similarly, the model is stable across different temperature coefficients $\tau$, with $\tau=0.2$ consistently performing well across datasets like Photo and WikiCS.
>
> Sensitivity to $\lambda_1$:
>
> | dataset | 0.001 | 0.01 | 0.1 | 0.5 | 1   | 2   |
> | --- | --- | --- | --- | --- | --- | --- |
> | photo | 95.07 | 95.10 | 95.27 | 95.03 | 94.90 | 94.85 |
> | wikics | 80.22 | 80.21 | 80.11 | 79.90 | 79.46 | 79.86 |
>
> Sensitivity to $\lambda_2$:
>
> | dataset | 0.001 | 0.01 | 0.1 | 0.5 | 1   | 2   |
> | --- | --- | --- | --- | --- | --- | --- |
> | photo | 95.15 | 95.29 | 95.27 | 95.23 | 95.19 | 94.95 |
> | wikics | 80.01 | 80.00 | 80.11 | 80.25 | 80.17 | 79.98 |
>
> Sensitivity to Temperature $\tau$:
>
> | dataset | 0.05 | 0.1 | 0.2 | 0.5 | 1   |
> | --- | --- | --- | --- | --- | --- |
> | photo | 94.92 | 95.16 | 95.27 | 94.98 | 95.01 |
> | wikics | 79.96 | 80.09 | 80.11 | 79.99 | 80.06 |
>
> **Limitations**: We have included a computational complexity analysis. We will add a Limitations Section in the final version to discuss applicability to dynamic/temporal graphs and performance on extremely large-scale graphs requiring sampling.
>
> [1] Sun, Jiaqi, et al. "Feature expansion for graph neural networks." *International Conference on Machine Learning*. PMLR, 2023.
>
> [2] Wu, Felix, et al. "Simplifying graph convolutional networks." *International conference on machine learning*. Pmlr, 2019.
>
> [3] Xu, Keyulu, et al. "How powerful are graph neural networks?." *arXiv preprint arXiv:1810.00826* (2018).

---

> > ### Author Rebuttal · Reviewer_Tndk · 2026-04-04
> >
> > Appreciate the clarification, which has addressed my concerns.

---

### Official Review · Reviewer_YZwz · 2026-03-13

**Soundness:** 3
**Presentation:** 3
**Significance:** 3
**Originality:** 3
**Overall Recommendation:** 4
**Confidence:** 3

**Summary:**

This paper presents a design of MoE for graphs, where the router is regularized by pre-defined node groups and the final layer weights are adjusted by a confidence mechanism defined by the node class prediction logits. Experiments have been performed on node classification tasks to demonstrate the accuracy improvement of the proposed model, and the better expert utilization & distribution from training.

**Compliance With Llm Reviewing Policy:**

Affirmed.

**Final Justification:**

I appreciate the authors' detailed rebuttal. The responses addressed my main concerns, and I would like to keep my positive rating.

**Key Questions For Authors:**

See weakness above

**Strengths And Weaknesses:**

## Strengths

+ The overall approach is reasonable. It's motivated by common observation in the MoE literature that experts may not specialize to the data well and may need extra guidance to align the model and data.
+ The design overall makes sense, combining data-driven components (contrastive routing) and theorem-motivated design (confidence-based weights)
+ The experiments are comprehensive and showcase meaningful improvements.


## Weaknesses

- The contrastive routing strategy is based on intuitive heuristic that partitions the nodes. First, I'm not sure if the semantic group construction is too restrictive. Requiring both feature and structure groups to align can be restrictive. Nodes with one matching group could still share significant commonality. Forcing them to separate may make the training unstable. Second, what types of graphs are suitable for such semantic group definition? For example, I would imagine homophily and heterophily graphs may require distinct criteria for semantic group definition.
- Theorem 3.1 may provide limited practical guidance. First, it only provides directional guidance saying that the routing weights should positively / negatively relate with the loss. Moreover, the conclusion is quite straightforward: it's basically saying that "if a path is good in the sense that it can reduce loss, then we want to increase its weight compared with other paths". This is basically a rephrase of the basic MoE gating design philosophy behind "top-K" selection.
- In ablation study, Table 3 shows that the full model achieves the highest accuracy. However, the delta compared with "w/o conf" and "w/o route" is marginal.
- The MoE variants compared in the experiments section follow diverse expert design patterns (e.g., some use multiple GNN experts, some use GNN and MLP experts, and different baselines may have different sparsity). It would be more convincing to include a comparison of complexity (w.r.t. computation or active / total parameters) in addition to the accuracy numbers currently shown.
- Routing based on "confidence" of node class prediction has been explored in previous works like Mowst-GNN. It would be good to discuss the relation between the proposed confidence gating and the one in Mowst-GNN.

---

> ### Author Rebuttal · Authors · 2026-03-29
>
> We appreciate the reviewer’s positive feedback. We address your concerns below:
>
>  **W1: About Semantic Group**
>
> Requiring both feature and semantic alignment is not restrictive, but necessary in graph settings. A node’s semantics are jointly determined by its attributes and structural context. Our semantic group construction aims to capture this joint consistency to provide discriminative supervision signals for the router. Specifically, considering only structure might group nodes with similar connectivity but distinct attributes, leading to pattern confusion. Relying solely on features might ignore crucial relational patterns. Besides, this mechanism is only used to regularize the router. It encourages nodes with dual-group alignments to exhibit consistent routing behaviors, while still preserving sufficient flexibility for nodes with one matching group to adapt dynamically during training.
>
> Empirically, we do not observe training instability. Instead, this guidance improves convergence and expert specialization.
>
> This work primarily focuses on homophily graphs. We agree that extending to heterophily graphs is an important direction, which may require alternative grouping criteria (e.g., role-based grouping). We will clarify this scope and discuss it as future work.
>
> **W2: Practical Value of Theorem 3.1**
>
> Theorem 3.1 is not a simple rephrase of the Top-K logic. Standard Top-K routing performs discrete selection and cannot quantify the relative reliability among selected experts — a gap our approach fills. Specifically, the theorem goes beyond merely demonstrate that optimal routing weights should be negatively correlated with loss. More crucially, through Theorem 3.1, Corollaries 3.2 and 3.3, we establish a theoretical foundation for using prediction confidence as a proxy to calibrate expert contributions. This calibration represents a novel theoretical insight that directly addresses the routing uncertainty that Top-K selection inherently ignores.
>
> **W3: Gains in Ablation Study**
>
> We conducted paired t-tests over 10 runs to evaluate the statistical significance. Experimental results show that our full model performs significantly better than the ablated variants (p < 0.05).
>
> **W4: Computational Complexity**
>
> We thank the reviewer for this important suggestion. We will include a detailed efficiency comparison in the final version.
>
> We evaluate complexity via FLOPs to demonstrate that C²GMoE maintains efficiency comparable to standard GNNs. For a standard GNN layer with hidden dimension $d_0$, the complexity is approximately $\mathcal{O}(|E|d_0 + Nd_0^2)$, where the feature transformation $Nd_0^2$ typically dominates the overhead. For a layer with $N$ nodes, $|E|$ edges, and $M$ total experts (where $1+k$ are active), we control the cost by scaling the hidden dimension $d = d_0 / \sqrt{1+k}$ relative to a standard GNN's dimension $d_0$.
>
> Expert Computation: The feature transformation cost, which typically dominates GNN overhead, becomes $(1+k) \cdot N(d_0/\sqrt{1+k})^2 = Nd_0^2$, aligning with standard GNNs. The neighbor aggregation cost becomes $\sqrt{1+k}|E|d_0$, which grows only sub-linearly and remains efficient for small $k$.
>
> Auxiliary Overhead: Additional modules such as the gating network $\mathcal{O}(NdM)$, contrastive routing $\mathcal{O}(Ngd)$, and confidence fusion $\mathcal{O}(Nd)$ all possess linear complexity w.r.t. $N$. These node-level operations incur FLOPs significantly lower than the core message-passing bottleneck.
>
> During inference, the contrastive routing module is discarded, ensuring **the total FLOPs remain on the same order of magnitude as standard GNNs** while providing significantly higher model capacity.
>
> We empirically evaluate the efficiency in the table below. The results show that C²GMoE maintains computational costs on the same order of magnitude as standard GNNs and Mowst, while outperforming DaMoE.
>
> |     | Total Params | Active Params | FLOPs | Inference Latency |
> | --- | --- | --- | --- | --- |
> | GCN | 388136 | 388136 | 5.90 G | $t_0$ |
> | +C²GMoE | 784900 | 396196 | 5.99 G | 1.82$t_0$ |
> | +Mowst | 582161 | 582161 | 8.86 G | 1.67$t_0$ |
> | +DaMoE | 1159623 | 772572 | 18.39 G | 3.19$t_0$ |
>
> **W5: Relation to Confidence-based Routing in Mowst-GNN**
>
> We will clarify it in the paper. C²GMoE differs from Mowst in two key aspects:
>
> - Motivation: Mowst employs confidence as a binary selection gate to dynamically choose between a GNN and an MLP. Its motivation is to decouple node feature modeling and graph structure modeling. Our method performs reliable fusion across multiple experts and is motivated to solve the widespread routing uncertainty problem in Graph-MoE systems.
>
> - Mechanism: Mowst relies on the logit dispersion of MLP outputs to estimate confidence without label supervision. We train a dedicated confidence predictor to explicitly estimate the probability of true label under label supervision, providing a more theoretically grounded weight for fusion.

---

> > ### Author Rebuttal · Reviewer_YZwz · 2026-04-02
> >
> > I appreciate the authors' detailed rebuttal. The responses addressed my main concerns, and I would like to keep my positive rating.

---

### Decision · Program_Chairs · 2026-04-30

**Decision:**

Accept (regular)

**Comment:**

This paper presents a Graph-MoE framework featuring Contrastive routing and Confidence-aware fusion. The significant novelty is the proposed group-wise contrastive routing strategy and confidence-aware fusion mechanism. Experiments are sufficient to support its statements. Three of the four reviewers provide positive feedback by recognizing the compelling empirical motivation, a new understanding of the existing methods, and solid technical contributions. Most concerns raised by the reviewers are properly resolved in the rebuttal.